# Spatial navigation as a digital marker for clinically differentiating cognitive impairment severity
Giorgio Colombo[1,2,9] ✉, Karolina Minta[1,3,9], William R. Taylor [1,4], Jascha Grübel[5], Eddie Chong[3,6], Joyce R. Chong[3,6], Mark J. H. Lim [3,6,7], Paul Nichol G. Gonzales[6], Mitchell K. P. Lai [3,6,7], Christopher P. Chen [3,6,7] & Victor R. Schinazi [1,8]

## Abstract

**Background** Spatial navigation impairments emerge early in Alzheimer's disease, but assessments targeting these deficits remain underutilised or impractical for cognitive screening. The Spatial Performance Assessment for Cognitive Evaluation (SPACE) is a newly developed digital tool that evaluates spatial navigation deficits associated with cognitive impairment.

**Methods** We assessed spatial navigation ability using SPACE in 300 older adults recruited from memory clinics and the general community. Participants were classified across different levels of cognitive impairment using the Clinical Dementia Rating (CDR) scale. Performance in SPACE was compared with clinical diagnosis, standard cognitive assessments, and demographic models using Area Under the ROC Curve (AUC), sensitivity, and specificity.

**Results** We show that SPACE reliably distinguishes CDR levels, exceeding the accuracy of demographic models and matching or surpassing most traditional neuropsychological tests. Including SPACE significantly increases the AUC for distinguishing between no dementia from mild dementia (0.76 to 0.94), no dementia from moderate dementia (0.79 to 0.95), and questionable dementia from mild dementia (0.70 to 0.91), all with consistently high sensitivity and specificity. A shortened version of SPACE, lasting less than 11 minutes, reduces administration time by 40% while maintaining high diagnostic accuracy. Cross-validation analyses confirm the reliability and robustness of these models.

**Conclusions** These findings highlight the potential of digital spatial navigation assessments to advance early detection, contributing to scalable and accessible healthcare.

## Plain language summary

Problems with spatial navigation ability, such as finding one's way around unfamiliar places, can appear early in Alzheimer's disease, but they are not often assessed in routine cognitive tests. This study examined a newly developed digital tool, the Spatial Performance Assessment for Cognitive Evaluation (SPACE), designed to measure these navigation difficulties. We tested SPACE in 300 individuals from memory clinics and the general community and compared it with clinical diagnosis and standard cognitive assessments. SPACE accurately distinguished between individuals with no dementia, mild dementia, and moderate dementia. A shorter version of SPACE (< 11 minutes) was also capable to distinguish between clinical diagnosis with high accuracy. These findings suggest that simple digital tests of spatial navigation ability could help detect cognitive impairment and make dementia screening more accessible and practical for the general population.

Dementia affects approximately 55 million people worldwide, and this number is projected to reach 150 million by 2050[1,2]. The global economic burden of dementia is estimated at $16.9 trillion, imposing substantial financial and social strain on individuals, families, and healthcare systems[3]. Alzheimer's Disease (AD) is the leading cause of dementia and is

characterised by progressive cognitive decline. AD typically unfolds across distinct stages, from Mild Cognitive Impairment (MCI)[4], where subtle memory and navigation difficulties emerge, to moderate and severe stages characterised by profound impairments in multiple cognitive domains, including memory, spatial orientation, and daily functioning[2,5–9]. Identifying

[1]Future Health Technologies, Singapore-ETH Centre, Campus for Research Excellence And Technological Enterprise (CREATE), Singapore, Singapore. [2]Chair of Cognitive Science, ETH Zurich, Zürich, Switzerland. [3]Department of Pharmacology, Yong Loo Lin School of Medicine, National University of Singapore, Singapore, Singapore. [4]Institute for Biomechanics, Department of Health Sciences and Technology, ETH Zurich, Zürich, Switzerland. [5]Department of Network and Data Science, Central European University, Vienna, Austria. [6]Memory, Aging and Cognition Centre, National University Health System, Singapore, Singapore. [7]Healthy Longevity Translational Research Programme, Yong Loo Lin School of Medicine, National University of Singapore, Singapore, Singapore. [8]Department of Psychology, Bond University, Gold Coast, QLD, Australia. [9]These authors contributed equally: Giorgio Colombo, Karolina Minta. ✉e-mail: gicolombo@ethz.ch

individuals with MCI provides an opportunity to delay disease progression through targeted interventions[8–11].

Episodic memory impairments have long been the hallmark of MCI and AD, forming the cornerstone of diagnostic criteria. Current neuropsychological assessments typically focus on the detection of memory deficits alongside impairments in other cognitive domains (e.g., attention, executive function). However, the limited sensitivity of some standard screening tools frequently hinders the detection of early or subtle cognitive changes, particularly in the preclinical stages of neurodegenerative disease[11–16]. As a result, many diagnoses are still made only after significant pathologic processes has already occurred, when interventions are limited. While blood biomarkers simplify early detection compared to Positron Emission Tomography (PET) and Cerebrospinal Fluid (CSF) methods, their reliance on clinic-based sampling and interpretation does not fully overcome the constraints of late-stage, in-clinic diagnosis.

The early stages of AD are characterised by neurodegenerative changes in the hippocampus and the entorhinal cortex, including neuronal loss as well as the accumulation of amyloid-beta (Aβ) peptides and phosphorylated tau (p-tau) protein[17–21]. Critically, these subcortical regions also play an essential role in coding spatial information in the environment[22] and in assisting individuals to keep track of changes in position and orientation during navigation[23]. Indeed, an accumulating body of evidence from real-world[24–26] and Virtual Reality (VR)[27–32] studies has identified spatial navigation deficits as a promising marker of genetic risk for sporadic AD (APOE ε4-carriers)[33–35] and a reliable means of distinguishing individuals across the spectrum of cognitive impairment[16,36]. Here, experimental paradigms that require individuals to learn and recall the configuration of landmarks in an environment (i.e. cognitive mapping)[24,31,37], maintain orientation and track self-motion (i.e. path integration)[27,28,32,38], as well as infer landmark positions from different viewpoints (i.e. perspective taking)[39,40] have all exhibited sensitivity to early cognitive impairment. Among these tasks, path integration deficits have been consistently associated with hippocampal[24,31,37] and entorhinal cortex[28,34] atrophy, along with reduced activity in these brain areas[33,34].

Advances in digital technologies have enabled the development of scalable, gamified, spatial navigation assessments that offer engaging[41], ecologically valid[42], and accessible tools for early cognitive screening[43–45]. Such digital platforms offer remote deployment, reducing reliance on specialised clinical settings and costly visits, while enabling continuous monitoring of disease progression[8,9,46,47]. Despite strong theoretical and empirical support for spatial navigation as an early marker of cognitive decline, most existing tools have yet to be validated against a gold standard diagnosis. As a result, few, if any, digital navigation tools have transitioned into clinical practice, representing a missed opportunity for early detection and intervention.

In this study, we administer the Spatial Performance Assessment for Cognitive Evaluation (SPACE)[48] to 300 individuals referred from memory clinics and the community and compare their performance with the Clinical Dementia Rating (CDR) scale and a battery of neuropsychological tests. SPACE assesses spatial navigation abilities through five tasks (i.e., path integration, egocentric pointing, mapping, associative memory, and perspective taking) on an iPad. The different tasks in SPACE require participants to navigate a virtual environment, keep track of their position and orientation, reconstruct spatial layouts, recall object-location associations, and estimate spatial relationships from various perspectives. We show that SPACE, can accurately discriminate between participants with various levels of cognitive impairment, as independently assessed by clinicians.

## Methods
### Participants
The study included 300 participants ($M_{age}$ = 74 years; 41% male). Patients were recruited from memory clinics at the National University Hospital (NUH) and St. Luke's Hospital in Singapore, while control participants were recruited from these sites (90%) and the community (10%). Participants

recruited from memory clinics were part of ongoing cohorts (HARMONISATION[49] and SINGER[50]) at the Memory, Aging & Cognition Centre (MACC). All recruited participants underwent an initial telephone screening to assess their eligibility for the study. Participants were eligible if they were over 50 years old and could walk 10 meters without using a walking aid. Participants were excluded if they manifested severe visual impairment or hearing loss, a history of seizure, epilepsy, or acute cardiac events. Signed informed consent was obtained from all participants or their legal representative prior to undergoing the experimental procedure. Ethical approval for this study was provided by the National Healthcare Group (NHG) Domain Specific Review Board (DSRB) in Singapore (2021/01160).

### Sociodemographic and health questionnaire
Participants completed a sociodemographic and health questionnaire that included questions on their age, gender, education level, handedness, tablet experience, previous navigation training, vision impairments, chronic health conditions, history of head trauma and falls, depression, anxiety and stress, smoking status, alcohol consumption, sleep, and physical activity. Furthermore, participants were asked to self-evaluate their spatial navigation abilities using the Santa Barbara Sense of Direction (SBSOD) scale[51].

### Global clinical dementia rating diagnosis
The global Clinical Dementia Rating (CDR) score was determined relative to six cognitive and functional domains: memory, orientation, judgment and problem solving, community affairs, home and hobbies, and personal care. The CDR was calculated using the CDR-assignment algorithm[52]. The CDR for each participant was either obtained from existing study records conducted within one year of the SPACE visit or reconstructed from comprehensive clinical notes and neuropsychological data when these records were unavailable. This method has been shown to achieve good to excellent agreement (intraclass correlation coefficient = 0.81–0.92) with face-to-face assessments[53]. CDR scores were used to classify impairment severity: 0 = no dementia, 0.5 = questionable dementia, 1 = mild dementia, 2 = moderate dementia, and 3 = severe dementia. Due to the limited number of participants with a CDR score of 3 ($n$ = 3), the moderate and severe groups were merged for analysis.

### Clinical consensus dementia diagnosis
In addition to CDR classification, patients received a diagnosis by clinical consensus based on MACC's procedures, which consider performance on the Vascular Dementia Battery (VDB), structural MRI findings, and relevant biomarkers. The VDB evaluates six cognitive domains (i.e. attention, language, verbal and visual memory, visuoconstruction, and visuomotor speed) and classifies domain impairment as "borderline" (failure in one domain-specific test but <50% of tasks impaired) or "impaired" (≥ 50% of tasks failed). This battery has been validated in Singaporean stroke and dementia cohorts[54]. Control participants had Mini-Mental State Examination (MMSE) scores of ≥ 23 (secondary or tertiary education) or ≥ 21 (primary or no education), and no cognitive domain impairment on the VDB. The diagnosis classified participants into groups reflecting cognitive severity (NCI = No Cognitive Impairment, SCD/SCI = Subjective Cognitive Decline/Impairment, MCI = Mild Cognitive Impairment, and Dementia), following the recommendation of the National Institute on Aging and the Alzheimer's Association (NIA-AA) for the diagnosis and evaluation of Alzheimer's Disease[55].

Etiological classification followed the recommendations of the NIA–AA and was supported by plasma and neuroimaging biomarkers. Plasma p-tau217 levels were quantified via SIMOA immunoassay and used to stratify participants into low (≤ 0.388 pg/mL), intermediate (0.387–0.470 pg/mL), or high (≥ 0.471 pg/mL) risk for PET positivity for brain amyloid pathology, according to thresholds validated in Singaporean cohorts[56,57]. A subset of participants ($n$ = 209) underwent multimodal MRI, with cerebrovascular disease (CeVD) defined as cortical infarcts and/or ≥2 lacunes and/or confluent white matter lesions in ≥2 regions (ARWMC ≥ 8)[58]. Accordingly, the cohort included individuals with AD

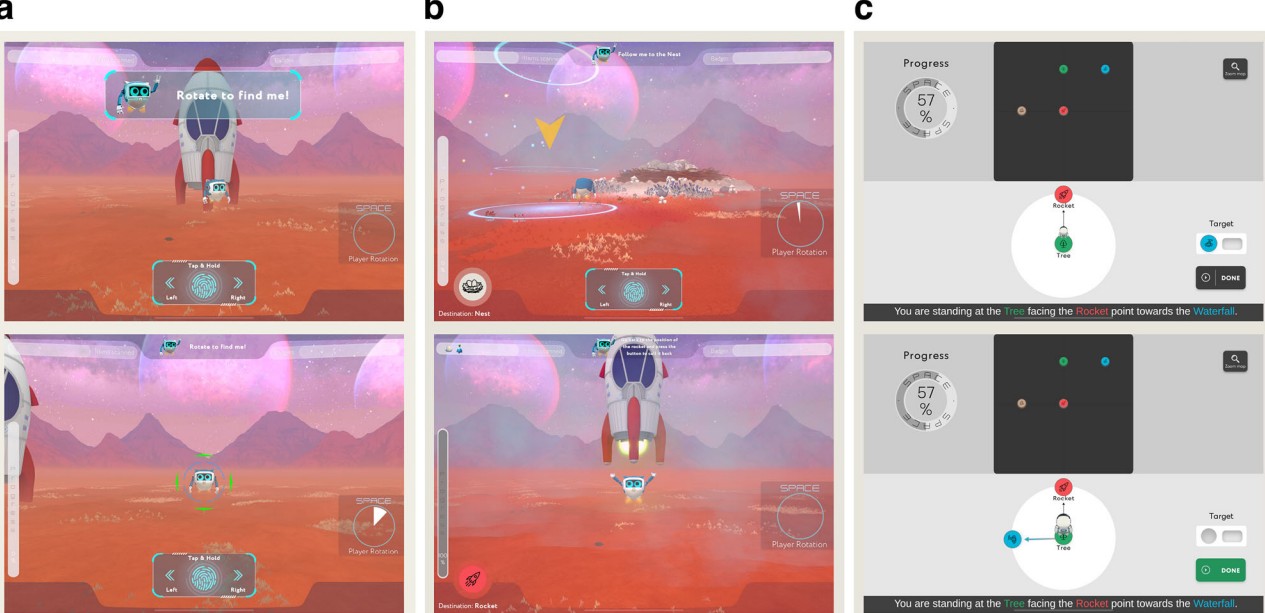

**Fig. 1 | Screenshots from SPACE tasks.** Start (top images) and end (bottom images) screens from three core tasks within the SPACE assessment. **a** Training phase, rotation: Participants rotate their viewpoint by swiping left or right on the screen to find the robot. **b** Path integration: Starting from the rocket and following the robot to two landmarks, participants must navigate back to the starting location without assistance. **c** Perspective taking: Participants view a top-down map of the environment and are asked to identify the direction of a landmark by imagining themselves standing at a specified location while facing a given orientation (e.g., "imagine you are standing at the tree, facing the rocket point to the waterfall).

Non-Vascular, AD Vascular, Non-AD Non-Vascular, and Non-AD Vascular pathologies – capturing the pathological diversity encountered in clinical settings. For participants without research scans, medical history (including CT scans when available) was used to determine CeVD status.

## Spatial performance assessment for cognitive evaluation (SPACE)

SPACE is a serious game deployed on the iPad to assess spatial navigation abilities that may be indicative of cognitive impairment. In SPACE, participants assume the role of an astronaut tasked with exploring and learning the positions of different landmarks on a planet while completing a series of spatial navigation tasks (Fig. 1). A comprehensive description of SPACE is provided in a previous publication[48].

SPACE begins with a training phase in which participants complete a series of manoeuvres designed to familiarise them with the controls while also assessing basic visuospatial skills. During this phase, participants learn how to rotate, move, and follow a robot from their starting point (rocket) to various destinations and back (Fig. 1a). Following the training phase, participants complete five spatial tasks that target different aspects of navigation. The path integration task evaluates how well participants can track changes in position and orientation as they navigate from the rocket to three landmarks. In each trial, participants follow the robot along two legs of a triangle and are then asked to return to the rocket via the third leg. At each landmark, the robot scans an item that participants will later be asked to recall (Fig. 1b). The pointing task measures how accurately participants can indicate the direction of other landmarks from a first-person perspective. Over a series of trials, participants are placed in front of a landmark and asked to indicate the direction of other landmarks in the environment. The mapping task assesses their ability to form a top-down mental map of the environment by reconstructing the spatial layout of landmarks. The associative memory task tests their recall of the specific items scanned at each landmark during the path integration task. Finally, the perspective taking task measures participants' ability to calculate the spatial relation between the landmarks from various perspectives. Here, the player is asked to imagine standing at a landmark facing another landmark by looking at the map of the planet. Participants are then required to indicate the correct bearing towards a third landmark (Fig. 1c).

Performance in SPACE is quantified using the following measures. Training time is the duration (in seconds) required for participants to complete the entire training task. Path integration distance error refers to the mean distance between the player's final position and the rocket's original position across all path integration trials. Pointing error is the mean absolute angular deviation (in degrees) from a starting position to the target location. Mapping accuracy is determined using bidimensional regression[59,60], a statistical method for assessing the degree of association ($R^2$) between the reference map and the map generated by the participants. The associative memory score is the percentage of correct pairings between scanned items and corresponding landmarks. Perspective taking error is the mean absolute angular deviation (in degrees) between the participant's directional estimate and the target landmark across all trials.

## Study procedure

Upon arriving at the study site, participants received an overview of the study procedures. Once written informed consent was obtained, participants completed a series of questionnaires to collect socio-demographic data, health information, and self-reported navigation ability. Participants then completed a battery of neuropsychological tests and SPACE. The neuropsychological tests included the MoCA, QDRS, Trail Making Test A and B (TMT-A/B), Maze Task, Animal Fluency, Digit Cancellation Test (DCT), and a Dual Task measure of divided attention. After completing SPACE, blood ($n = 300$) and saliva samples ($n = 96$) were collected for biomarker analyses. Participants also completed a short gait assessment using single and dual tasks. The entire testing procedure lasted approximately 3 hours. This study is part of a wider data collection effort to validate SPACE in community and clinical samples.

## Statistics and reproducibility

All statistical analyses were considered statistically significant at $p$-values < 0.05. Two-sided tests and Bonferroni corrections were applied whenever

appropriate. All analyses and plots were performed using Jamovi (v.2.6.26.0) and RStudio (v.2024.12.0 + 467). Descriptive statistics were computed for key demographic, clinical, neuropsychological, and task-performance variables across CDR levels. Given violations of the assumptions of normality of residuals and homogeneity of variance, we examined group differences in performance on the spatial navigation tasks of SPACE using robust one-way ANOVAs (modified one-step estimator; 5,000 bootstrap samples). To account for age and gender differences in navigation performance, we additionally conducted ANCOVAs for each SPACE task with these variables included as covariates. As the ANCOVAs supported the group differences reported with the robust ANOVAs, we retained the robust ANOVA as our primary approach. Robust ANOVAs provide unbiased estimates under violations of normality, homogeneity of variance, and unequal group sizes[61], which was the case for most of our models. We report the results of the ANCOVA in the Supplementary Note 7 for transparency and comparison.

Logistic regressions were then used to model the relationship between predictors and all combinations of binary outcomes from the CDR scale. Following the methodology from previous researchers[44], we compared three model types: a demographic model with age, education and gender (*Dem*), a model including only the significant tasks in SPACE (*SPACE*), and a combined model including both demographic variables and the SPACE tasks (*Total*). The purpose of this comparison was to establish a baseline model using known demographic risk factors for dementia, against which the added discriminative value of SPACE performance could be evaluated. Model performance was compared using DeLong's test to evaluate the incremental contribution of SPACE beyond demographic predictors. The optimal classification cutoff for each model was determined using the Youden index and assessed using the AUC, sensitivity, specificity, and accuracy. Model performance was further evaluated using deviance, AIC, and the Brier score. Given that only two participants with a CDR of 2+ completed the path integration, pointing, and mapping tasks, this group were excluded from the analysis for these tasks. To account for the results of the ANCOVA, we additionally evaluated a SPACE model that included the pointing task alongside demographic variables (Supplementary Table 9).

We cross-validated our models using k-fold cross-validation and Leave-One-Out Cross-Validation (LOOCV). For both methods, we provide estimates of model performance, including AUC, sensitivity, and specificity, along with their corresponding confidence intervals. The k-fold cross-validation involved partitioning the dataset into k subsets (k = 10), training the model on k-1 subsets, and testing it on the remaining subset. LOOCV involves training the model on all but one data point and testing it on the remaining point, iterating this process for each data point in the dataset.

To compare the performance of SPACE relative to traditional neuropsychological tests, we conducted ROC analyses for each CDR stage contrast. Differences in AUC values between SPACE and other neuropsychological tests were examined using the DeLong test with Bonferroni correction. The same analysis was conducted to assess whether SPACE can discriminate between categories from the consensus diagnosis (NCI vs. Dementia, NCI vs. MCI, and MCI vs. Dementia). For these analyses, we merged the No Cognitive Impairment (NCI) and Subjective Cognitive Decline/Impairment (SCD/SCI) groups into a single NCI category, as SCD/SCI reflects concerns about cognitive changes but does not constitute a formal clinical diagnosis of cognitive impairment. Finally, we evaluated whether a short version of SPACE, excluding the path integration task, could discriminate between different CDR levels. Here, a Repeated Measures ANOVA was employed to assess differences in the time taken to complete the three tasks (training, path integration, and perspective taking). Logistic regressions, ROC curve analysis, and cross-validation (stratified k-fold and LOOCV) were also performed on the short version of SPACE.

Since the primary focus of this paper is to assess the performance of SPACE relative to the CDR scores, various other measures were excluded from the main analyses. A complete list of all measures collected and additional analyses is reported in Supplementary Notes 1–10.

## Results

### Group differences in spatial navigation performance

Tables 1–3 summarise cohort demographics, clinical characteristics, and performance on neuropsychological tests and SPACE across CDR levels. Due to the very small number of participants (*n* = 3) with a CDR score of 3, the CDR 2 and CDR 3 groups were combined for analysis (see Supplementary Note 2). We performed robust ANOVAs to assess differences in spatial navigation performance across CDR groups (CDR 0, CDR 0.5, CDR 1, and CDR 2+) for each task in SPACE (Fig. 2 and Supplementary Table 2). Given that only two participants with a CDR of 2+ completed the path integration, pointing, and mapping tasks, this group were excluded from the analysis for these tasks. Results revealed significant CDR group differences in training time ($F_{(3, 296)} = 14.7$, $p < 0.001$, $\xi = 0.657$), path integration distance error ($F_{(3, 296)} = 7.0$, $p = 0.004$, $\xi = 0.587$), and perspective taking error ($F_{(3, 296)} = 11.0$, $p < 0.001$, $\xi = 0.51$). No significant group differences were observed for egocentric pointing error ($F_{(3, 296)} = 2.61$, $p = 0.101$) or mapping accuracy ($F_{(3, 296)} = 1.33$, $p = 0.291$).

Post-hoc comparisons revealed that participants with CDR 0 completed the training phase significantly faster than those with CDR 0.5 ($p = 0.022$, $\xi = 0.229$), CDR 1 ($p < 0.001$, $\xi = 0.717$), and CDR 2+ ($p < 0.001$, $\xi = 0.886$). The CDR 0.5 group was also significantly faster than both the CDR 1 ($p = 0.001$, $\xi = 0.419$) and CDR 2+ ($p < 0.001$, $\xi = 0.761$) groups. Finally, participants with CDR 1 were faster than those with CDR 2+ ($p = 0.022$, $\xi = 0.483$). For path integration distance error, participants with CDR 0 performed significantly better than those with CDR 1 ($p < 0.001$, $\xi = 0.666$), but not better than those with CDR 0.5 ($p = 0.44$). Participants with CDR 0.5 also outperformed those with CDR 1 ($p < 0.001$, $\xi = 0.714$). For perspective taking error, participants with CDR 0 performed significantly better than those with CDR 1 ($p < 0.001$, $\xi = 0.496$) and CDR 2+ ($p < 0.001$, $\xi = 0.661$) but did not differ from those with CDR 0.5 ($p = 0.572$). Participants with CDR 0.5 also outperformed those with CDR 1 ($p < 0.001$, $\xi = 0.518$) and CDR 2+ ($p < 0.001$, $\xi = 0.613$). No differences emerged between participants with CDR 1 and CDR 2+ ($p = 0.476$).

### Diagnostic classification performance

We conducted logistic regression analyses to evaluate the diagnostic utility of the SPACE measures (i.e. training, path integration, and perspective taking), which showed significant effects in the previous ANOVAs. These analyses aimed to classify participants across CDR stages by comparing between diagnostic groups. Here, we compared baseline models that included demographic variables (i.e. age, education, and gender) to models that additionally incorporated the significant SPACE metrics. We assessed improvements in model fit using likelihood-ratio tests (see Table 4 and Supplementary Table 3 for full model statistics) and evaluated classification accuracy using the Area Under the Curve (AUC), sensitivity, and specificity.

Incorporating SPACE measures improved model performance for all diagnostic contrasts except for CDR 0 *vs.* 0.5 (Fig. 3). Importantly, these measures improved classification not only for CDR 2+ but also for early stages, including CDR 0.5 and CDR 1 (Table 4 & 5). Comparison between CDR 0 and CDR 0.5 produced the smallest improvements. Here, the demographic model yielded an AUC of 0.57, with sensitivity of 0.40 and specificity of 0.75. Adding SPACE raised the AUC slightly to 0.61 ($p_{DeLong} = 0.256$), sensitivity to 0.47 and specificity to 0.79, but this change was not statistically significant ($\Delta\chi^2 = 4.88$, $p = 0.181$). Greater improvements were achieved for all the other five models. As expected, the addition of SPACE to contrasts involving the CDR 2+ group resulted in considerable gains in diagnostic performance. More importantly, however, for the comparison between CDR 0 and CDR 1, the demographic model achieved an AUC of 0.76, with sensitivity of 0.74 and specificity of 0.71. Critically, adding SPACE significantly improved model fit ($\Delta\chi^2 = 34.7$, $p < 0.001$), increasing the AUC to 0.94 ($p_{DeLong} < 0.001$), sensitivity to 1.00, and specificity to 0.85. Of similar clinical importance was the differentiation between CDR 0.5 from CDR 1, where the demographic model produced an AUC of 0.70, with sensitivity of 0.74 and specificity of 0.59. Again, the

**Table 1 | Demographic characteristics**

| Clinical dementia rating (CDR) | | | | | | | | |
|---|---|---|---|---|---|---|---|---|
| Characterisation | Variable | CDR = 0, N = 153 (51%) | CDR = 0.5, N = 96 (32%) | CDR = 1, N = 34 (11%) | CDR = 2, N = 14 (5%) | CDR = 3, N = 3 (1%) | Overall, N = 300 | p-value |
| **Demographic** | Age, years | 74 (69, 78) | 75 (70, 78) | 75 (69, 78) | 78 (73, 79) | 87 (81, 88) | 74 (70, 78) | 0.079 |
| | Gender, n (%) | | | | | | | 0.411 |
| | Male | 63 (41%) | 40 (42%) | 17 (50%) | 4 (29%) | 0 (0%) | 124 (41%) | |
| | Female | 90 (59%) | 56 (58%) | 17 (50%) | 10 (71%) | 3 (100%) | 176 (59%) | |
| | Education level, n (%) | | | | | | | 0.010 |
| | No formal education | 6 (3.9%) | 5 (5.2%) | 0 (0%) | 2 (14%) | 0 (0%) | 13 (4.3%) | |
| | Primary school | 13 (8.5%) | 12 (13%) | 10 (29%) | 3 (21%) | 1 (33%) | 39 (13%) | |
| | Secondary school | 45 (29%) | 32 (33%) | 15 (44%) | 7 (50%) | 1 (33%) | 100 (33%) | |
| | High school | 35 (23%) | 24 (25%) | 6 (18%) | 2 (14%) | 1 (33%) | 68 (23%) | |
| | University | 54 (35%) | 23 (24%) | 3 (8.8%) | 0 (0%) | 0 (0%) | 80 (27%) | |
| | Depression, score | 1.00 (1.00, 2.00) | 1.00 (1.00, 4.00) | 1.00 (1.00, 1.75) | 1.00 (1.00, 1.75) | 1.00 (1.00, 3.00) | 1.00 (1.00, 2.13) | 0.097 |
| | Anxiety, score | 2.00 (1.00, 3.00) | 2.00 (1.00, 3.63) | 1.00 (1.00, 2.00) | 1.00 (1.00, 2.00) | 1.00 (1.00, 2.50) | 1.00 (1.00, 3.00) | 0.140 |
| | Stress, score | 1.00 (1.00, 3.00) | 1.00 (1.00, 3.25) | 1.00 (1.00, 2.00) | 1.00 (1.00, 2.75) | 1.00 (1.00, 2.00) | 1.00 (1.00, 3.00) | 0.552 |
| | Sleep, hours | 6.50 (5.50, 7.50) | 7.00 (5.50, 8.00) | 8.00 (7.00, 9.75) | 9.25 (8.00, 10.00) | 9.00 (8.50, 10.50) | 7.00 (6.00, 8.00) | <0.001 |
| | Tablet experience, n (%) | | | | | | | <0.001 |
| | None | 12 (7.8%) | 24 (25%) | 13 (38%) | 10 (71%) | 2 (67%) | 61 (20%) | |
| | Low | 48 (31%) | 31 (32%) | 17 (50%) | 4 (29%) | 1 (33%) | 101 (34%) | |
| | High | 93 (61%) | 41 (43%) | 4 (12%) | 0 (0%) | 0 (0%) | 138 (46%) | |
| | SBSOD, score | 3.57 (2.79, 4.36) | 3.93 (3.13, 4.64) | 3.75 (3.16, 4.21) | 4.46 (3.59, 5.48) | 6.86 (6.14, 6.93) | 3.71 (2.93, 4.50) | 0.001 |

For continuous variables we report the median (IQR). For categorial variables we report the raw number (%). The p-value was calculated using the two-sided Kruskal-Wallis rank sum test or Pearson's Chi-squared test. Supplementary Table 1 details duration of dementia.
Demographic characteristics of the sample stratified by CDR status.

addition of SPACE significantly improved the model ($\Delta\chi^2 = 32.9, p < 0.001$), increasing the AUC to 0.91 ($p_{DeLong}$ <0.001), sensitivity to 0.95 and specificity to 0.73.

**Cross-validation performance**
To further evaluate the classification performance of SPACE, we performed stratified k-fold cross-validation (k = 10) and Leave-One-Out Cross-Validation (LOOCV) for each of the contrasts (Table 6). In both analyses, SPACE retained high diagnostic accuracy across most group comparisons. As expected, classification between CDR 0 and CDR 0.5 produced the weakest results. However, performance remained strong in all other clinically relevant contrasts. For distinguishing CDR 0 from CDR 1, k-fold cross-validation produced an AUC of 0.87 with sensitivity of 0.79 and specificity of 0.83. LOOCV yielded similar results with an AUC of 0.85, sensitivity of 0.79, and specificity of 0.82. For the comparison between CDR 0.5 and CDR 1, k-fold cross-validation produced an AUC of 0.80, with sensitivity of 0.74, and specificity of 0.75. LOOCV obtained an identical AUC of 0.80, sensitivity of 0.63, and specificity of 0.75. In the comparison between CDR 0 and CDR 2 +, k-fold cross-validation and LOOCV reached AUCs of 0.89 and 0.88 respectively, with sensitivity higher than 0.69 and specificity above 0.86 in both cases.

**Neuropsychological tests**
To compare the diagnostic accuracy of SPACE relative to standard neuropsychological measures, ROC analyses were performed, and differences in AUCs were assessed using Bonferroni-corrected DeLong test (see Fig. 4 and Supplementary Table 4). For CDR 0 vs 0.5, no neuropsychological test differed significantly from SPACE, and all showed modest accuracy (AUC ≈ 0.60–0.73). When comparing CDR 0 vs 1, MoCA outperformed SPACE (p = 0.021), whereas the QDRS (p = 0.010), Maze (p = 0.010), and Dual Task (p = 0.007) performed significantly worse. In differentiating CDR 0.5 vs 1, SPACE again achieved high discrimination accuracy (AUC = 0.91)

and was significantly better than the QDRS (p = 0.003), TMT-A (p = 0.025), Maze (p = 0.008), and Dual Task (p = 0.004). No significant differences were observed for CDR 0 vs 2 +, 0.5 vs 2 +, or 1 vs 2 +, where all tasks, including SPACE, performed strongly (AUCs > 0.85). Overall, SPACE showed the same or better discrimination than most conventional tests, particularly for transitions from very mild to mild dementia, with AUCs above 0.90 in all but the earliest (CDR 0 vs 0.5) and most advanced (CDR 1 vs 2 +) stage comparisons.

While the primary goal of this paper was to assess whether SPACE can discriminate between various CDR scores, we also compared it relative to consensus diagnosis. For each comparison, we calculated AUC values along with sensitivity and specificity at the optimal cut-off. The p-values reported in the table indicate the statistical tests comparing the AUC of each assessment with that of SPACE, which served as the reference model. For these analyses, we merged the No Cognitive Impairment (NCI) and Subjective Cognitive Decline/Impairment (SCD/SCI) groups into a single NCI category, as SCD/SCI reflects concerns about cognitive changes but does not constitute a formal clinical diagnosis of cognitive impairment. In distinguishing between NCI from Dementia, SPACE demonstrated excellent discrimination accuracy (AUC = 0.94). In this comparison, MoCA performed significantly better than SPACE (AUC = 1.00, p = 0.006), whereas QDRS performed significantly worse (AUC = 0.82, p = 0.037). All other tests showed no significant difference from SPACE. For NCI versus MCI, MoCA again outperformed SPACE (AUC 0.86 vs AUC 0.72, p = 0.002), while the remaining assessments were statistically indistinguishable from it. Finally, for MCI versus Dementia, none of the cognitive measures differed significantly from SPACE (see Supplementary Tables 5 and 6).

**Short SPACE**
On average, participants took 281 seconds to complete the SPACE training, 465 seconds for path integration, and 386 seconds for perspective taking tasks. A repeated measures ANOVA revealed significant differences in time

**Table 2 | Clinical and neuropsychological characteristics. Clinical diagnoses and neuropsychological performance stratified by CDR status**

| Characterisation | Variable | Clinical Dementia Rating (CDR) | | | | | | |
| --- | --- | --- | --- | --- | --- | --- | --- | --- |
| | | CDR = 0, N = 153 (51%) | CDR = 0.5, N = 96 (32%) | CDR = 1, N = 34 (11%) | CDR = 2, N = 14 (5%) | CDR = 3, N = 3 (1%) | Overall, N = 300 | p-value |
| **Clinical** | Clinical diagnosis, n (%) | | | | | | | <0.001 |
| | NCI | 86 (56%) | 21 (22%) | 0 (0%) | 0 (0%) | 0 (0%) | 107 (36%) | |
| | SCD/SCI | 50 (33%) | 17 (18%) | 0 (0%) | 0 (0%) | 0 (0%) | 67 (22%) | |
| | MCI | 17 (11%) | 54 (56%) | 2 (5.9%) | 0 (0%) | 0 (0%) | 73 (24%) | |
| | Dementia | 0 (0%) | 4 (4.2%) | 32 (94%) | 14 (100%) | 3 (100%) | 53 (18%) | |
| | Aetiology, n (%) | | | | | | | <0.001 |
| | AD, Non-Vascular | 17 (11%) | 15 (16%) | 18 (53%) | 5 (36%) | 1 (33%) | 56 (19%) | |
| | AD, Vascular | 3 (2.0%) | 7 (7.3%) | 5 (15%) | 8 (57%) | 2 (67%) | 25 (8.3%) | |
| | Non-AD, Non-Vascular | 103 (67%) | 52 (54%) | 5 (15%) | 0 (0%) | 0 (0%) | 160 (53%) | |
| | Non-AD, Vascular | 30 (20%) | 22 (23%) | 6 (18%) | 1 (7.1%) | 0 (0%) | 59 (20%) | |
| | Significant CeVD status, n (%) | | | | | | | 0.004 |
| | No | 120 (78%) | 67 (70%) | 23 (68%) | 5 (36%) | 1 (33%) | 216 (72%) | |
| | Yes | 33 (22%) | 29 (30%) | 11 (32%) | 9 (64%) | 2 (67%) | 84 (28%) | |
| | pTau risk, n (%) | | | | | | | <0.001 |
| | High | 27 (18%) | 26 (27%) | 25 (74%) | 14 (100%) | 3 (100%) | 95 (32%) | |
| | Low | 126 (82%) | 70 (73%) | 9 (26%) | 0 (0%) | 0 (0%) | 205 (68%) | |
| | Genotype, n | | | | | | | 0.192 |
| | E2/E2 | 3 (6.1%) | 0 (0%) | 0 (0%) | 0 (0%) | 0 (0%) | 3 (3.1%) | |
| | E2/E3 | 5 (10%) | 2 (6.7%) | 1 (10%) | 1 (17%) | 0 (0%) | 9 (9.4%) | |
| | E3/E3 | 32 (65%) | 17 (57%) | 4 (40%) | 4 (67%) | 0 (0%) | 57 (59%) | |
| | E3/E4 | 9 (18%) | 10 (33%) | 5 (50%) | 0 (0%) | 1 (100%) | 25 (26%) | |
| | E4/E4 | 0 (0%) | 1 (3.3%) | 0 (0%) | 1 (17%) | 0 (0%) | 2 (2.1%) | |
| **Neuropsychological** | MoCA, score | 27.0 (26.0, 28.0) | 25.5 (20.8, 27.0) | 16.0 (12.3, 17.8) | 9.5 (7.3, 12.0) | 5.0 (3.0, 5.5) | 26.0 (20.0, 28.0) | <0.001 |
| | MoCA (education adj.), score | 28.0 (27.0, 29.0) | 26.0 (20.8, 27.0) | 16.0 (13.3, 18.0) | 10.5 (8.3, 12.8) | 6.0 (4.0, 6.0) | 27.0 (21.0, 28.0) | <0.001 |
| | QDRS informant, score | 0.50 (0.00, 1.50) | 1.50 (0.50, 3.00) | 2.00 (0.50, 3.50) | 0.75 (0.00, 1.13) | 2.00 (1.00, 3.00) | 1.00 (0.00, 2.25) | <0.001 |
| | Trail Making A, seconds | 35 (27, 47) | 44 (31, 64) | 70 (49, 90) | 150 (75, 150) | 150 (150, 150) | 41 (29, 66) | <0.001 |
| | Trail Making B, seconds | 89 (68, 136) | 117 (78, 261) | 300 (156, 300) | 300 (300, 300) | 300 (300, 300) | 106 (75, 240) | <0.001 |
| | Digit Cancellation Test, n | 24 (20, 29) | 21 (17, 28) | 17 (11, 20) | 9 (6, 12) | 3 (2, 12) | 22 (17, 28) | <0.001 |
| | Animal Fluency, n | 17 (15, 21) | 15 (11, 19) | 9 (6, 11) | 6 (3, 8) | 0 (0, 0) | 15 (11, 19) | <0.001 |
| | Dual task, n | 94 (86, 102) | 92 (85, 98) | 81 (72, 95) | 88 (70, 94) | 44 (22, 67) | 92 (84, 100) | <0.001 |
| | Maze task, seconds | 25 (18, 35) | 29 (22, 40) | 49 (27, 60) | 63 (33, 240) | 240 (240, 240) | 28 (21, 44) | <0.001 |

For continuous variables we report the median (IQR). For categorial variables we report the raw number (%). The p-value was calculated using the two-sided Kruskal-Wallis rank sum test or Pearson's Chi-squared test.

**Table 3 | SPACE task performance. Performance on the spatial navigation tasks in SPACE stratified by CDR status**

| Characteri-sation | Variable | CDR = 0, N = 153 (51%) | CDR = 0.5, N = 96 (32%) | CDR = 1, N = 34 (11%) | CDR = 2, N = 14 (5%) | CDR = 3, N = 3 (1%) | Overall, N = 300 | p-value |
| --- | --- | --- | --- | --- | --- | --- | --- | --- |
| **SPACE** | Training, seconds | 262 (238, 293) | 274 (245, 313) | 309 (282, 364) | 352 (318, 404) | 424 (376, 436) | 274 (246, 312) | <0.001 |
| | Path integration, meters | 196 (157, 261) | 212 (170, 240) | 290 (236, 372) | 308 (236, 379) | – | 205 (161, 273) | <0.001 |
| | Pointing, degrees | 76 (65, 89) | 78 (67, 94) | 92 (74, 102) | 102 (98, 106) | – | 79 (67, 92) | 0.016 |
| | Mapping, R² | 0.56 (0.30, 0.86) | 0.57 (0.24, 0.84) | 0.77 (0.40, 0.85) | 0.80 (0.80, 0.81) | – | 0.57 (0.27, 0.85) | 0.453 |
| | Memory, % correct | | | | | | | <0.001 |
| | 0 | 1 (0.7%) | 1 (1.2%) | 4 (20%) | 0 (0%) | – | 6 (2.4%) | |
| | 33 | 7 (4.7%) | 8 (9.6%) | 3 (15%) | 1 (50%) | – | 19 (7.5%) | |
| | 100 | 142 (95%) | 74 (89%) | 13 (65%) | 1 (50%) | – | 230 (90%) | |
| | Perspective taking, degrees | 42 (28, 58) | 43 (31, 57) | 58 (49, 67) | 60 (54, 72) | 58 (51, 66) | 46 (32, 61) | <0.001 |

For continuous variables we report the median (IQR). For categorial variables we report the raw number (%). The p-value was calculated using the two-sided Kruskal-Wallis rank sum test or Pearson's Chi-squared test.

**Fig. 2 | Between-group comparisons across CDR levels.** Robust ANOVA with Bonferroni-corrected pairwise comparisons. **a** Training time, (**b**) Path integration distance, and (**c**) Perspective taking error. Boxplots display group medians and inter-quartile ranges, with individual data points overlaid, *n* indicates the number of participants for each CDR group.

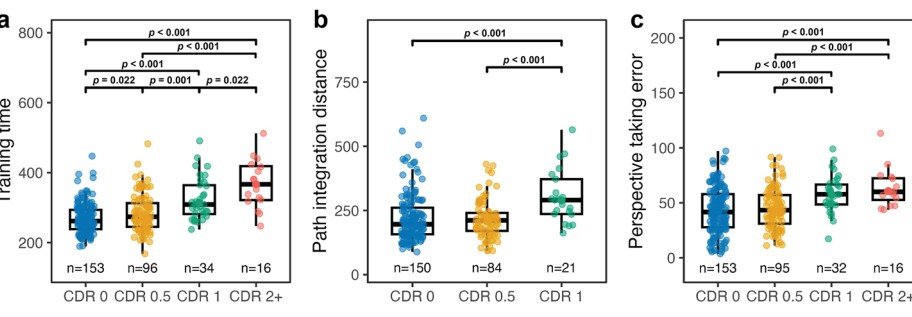

to completion for these tasks ($F_{(2, 498)} = 75.28$, $p < 0.001$, $\eta^2 = 0.232$). Participants completed the training phase significantly faster than both the path integration and perspective taking tasks. Additionally, participants were faster at completing the perspective taking task than the path integration task. All Bonferroni post-hoc comparisons were significant ($p < 0.001$).

To assess whether a more time-efficient and user-friendly version of SPACE could also effectively discriminate between CDR stages, we repeated the classification analyses using only the training phase and perspective taking task (Table 7). In this short version of SPACE (sSPACE), which required only an average of 667 seconds, the CDR 0 *vs.* CDR 0.5 contrast yielded an AUC of 0.61 (sensitivity = 0.49, specificity = 0.72). Again, the clinically relevant CDR 0 *vs.* CDR 1 contrast resulted in an AUC of 0.91, (0.91, 0.81), while the CDR 0.5 *vs.* CDR 1 comparison yielded an AUC of 0.82 (0.84, 0.78). Since the models with the moderate group did not include the path integration task, contrasts for sSPACE were limited to the CDR 0, CDR 0.5, and CDR 1 groups (Fig. 5).

Given the known role of path integration in discriminating impaired from not-impaired individuals, we compared AUCs for each CDR contrast in sSPACE with and without the path integration task. Here again, we excluded contrasts involving CDR 2+ because participants in these groups frequently aborted the path integration task due to its length and complexity, opting instead to proceed directly to the perspective taking task. The addition of the path integration task did not significantly improve discrimination accuracy for CDR 0 vs CDR 0.5 ($p = 0.777$, $p_{bonferroni} = 1.000$) and CDR 0 vs CDR 1 ($p = 0.107$, $p_{bonferroni} = 0.322$). We found that including the path integration task significantly improved the AUC for CDR 0.5 and CDR 1 ($p = 0.019$), increasing it from 0.83 to 0.91. However, this comparison did not survive Bonferroni correction ($p_{Bonferroni} = 0.056$).

## Discussion

We developed a tablet-based spatial navigation assessment (SPACE) and tested it in a clinical and community cohort of 300 individuals diagnosed with varying levels of cognitive impairment using the CDR scale. SPACE improved classification performance beyond demographic models across most group comparisons, including CDR 0 *vs.* CDR 1 and CDR 2+, as well as CDR 0.5 *vs.* CDR 1 and CDR 2+. Classification accuracy was consistently high across these comparisons, with AUCs ranging from 0.91 to 0.95, sensitivities from 0.88 to 1.00, and specificities from 0.73 to 0.88. SPACE maintained high accuracy even when distinguishing between adjacent early stages. We also evaluated a shorter version of SPACE (sSPACE), designed for rapid clinical screening and unsupervised deployment at home and in community settings. In under 11 minutes, sSPACE retained diagnostic accuracy comparable to the full version. Cross-validation analyses confirmed the reliability and robustness of these models, further highlighting their suitability for scalable cognitive screening. Altogether, these findings underscore the importance of incorporating spatial navigation tests into cognitive assessments and their sensitivity for detecting subtle differences in cognitive status.

While demographic models alone achieved modest classification performance, the inclusion of SPACE consistently improved model accuracy, sensitivity, and specificity across all diagnostic contrasts, although gains

were not significant when comparing CDR 0 *vs.* CDR 0.5 and CDR 1 *vs.* CDR 2+. As expected, SPACE increased the AUC for the model distinguishing CDR 0 *vs.* CDR 2+ (AUC increase from 0.79 to 0.95) and when distinguishing CDR 0.5 *vs.* CDR 2+ (AUC increase from 0.80 to 0.91). Across both of these contrasts, sensitivity and specificity remained high, ranging from 0.88 to 0.94 and 0.85 to 0.88, respectively. Notably, there was also a substantial improvement when differentiating CDR 0 *vs.* CDR 1 (AUC increase from 0.76 to 0.94) and CDR 0.5 *vs.* CDR 1 (AUC increase from 0.70 to 0.91). Here, we also observed substantial gains in both sensitivity and specificity. For the contrast between CDR 0 *vs.* CDR 1, sensitivity increased from 0.74 to 1.00 and specificity from 0.71 to 0.85. Similarly, for the contrast between CDR 0.5 *vs.* CDR. 1, sensitivity rose from 0.74 to 0.95 and specificity from 0.59 to 0.73.

SPACE classification performance for CDR 0 *vs.* CDR 0.5 and CDR 1 *vs.* CDR 2+ was modest. This result is likely due to a small overlap between the clinical classification of these disease stages, rather than a limitation of the navigation tests themselves[62]. The CDR classification between CDR 0.0 *vs.* 0.5 is based on the detection of subtle cognitive deficits, primarily relying on memory performance during the interview. In fact, previous work has sometimes sidestepped this challenge by simply combining the classifications of questionable dementia and mild dementia[63,64] or by completely omitting one of the categories[65]. However, even when addressing marginal comparisons, SPACE consistently improved model performance compared to sociodemographic variables alone. These findings were validated using stratified 10-fold cross-validation and LOOCV, with both methods yielding similarly high AUCs, sensitivity, and specificity across most contrasts. These results are also consistent with those from the consensus diagnosis. Apart from the MoCA, which performed exceedingly well in this cohort, SPACE demonstrated clear discriminative ability comparable to that of other neuropsychological tests. This convergence indicates that SPACE captures clinically meaningful differences in cognitive status across independent diagnostic methods.

Previous research using an unsupervised mobile app focusing on episodic memory demonstrated promising cross-validated performance in distinguishing mild cognitive impairment from unimpaired individuals in memory clinic settings, with an AUC of 0.83, increasing to 0.87 when based on multiple sessions[44]. Other researchers have shown that memory (AUC = 0.92) and orientation tasks (AUC = 0.91) can discriminate between AD and frontotemporal dementia[32], and that cognitive mapping tasks, rather than episodic memory, can predict disease progression (AUC = 0.8)[31]. Building on this evidence, we show that a tablet-based spatial navigation test battery is able to achieve comparable or superior classification accuracies (AUCs = 0.94–0.95), confirmed through cross-validation, both for detecting CDR 1 and CDR 2+ impairments. Importantly, deficits in spatial navigation appear to be an excellent marker for discretising subtle clinical stages such as those from questionable (CDR 0.5) to mild (CDR 1) impairments.

To address the challenges associated with typical, costly clinical assessments, it is critical that novel tools deliver efficient, scalable, and user-friendly digital solutions that can be deployed in community-based or other unsupervised screening contexts. By focusing on two of the most

**Table 4 | Model fit and classification performance.** Comparison of model fit and predictive accuracy for distinguishing clinical groups, with and without the inclusion of SPACE measures. Overall model comparisons were performed using likelihood-ratio chi-squared tests

| Model | Dev | AIC | Brier | MCC | R²N | Overall model Test | | | Predictive measures | | | | | | |
|---|---|---|---|---|---|---|---|---|---|---|---|---|---|---|---|
| | | | | | | χ² | df | p | AUC | Upp CI | Low CI | Opt Cutoff | Sens | Spec | Acc |
| **CDR 0 vs. CDR 0.5** | | | | | | | | | | | | | | | |
| Age, Edu, Gender | 300 | 314 | 0.23 | 0.14 | 0.02 | 3.53 | 6 | 0.739 | 0.57 | 0.50 | 0.64 | 0.39 | 0.40 | 0.75 | 0.62 |
| + SPACE | 295 | 315 | 0.22 | 0.27 | 0.05 | 8.41 | 9 | 0.493 | 0.61 | 0.53 | 0.69 | 0.40 | 0.47 | 0.79 | 0.67 |
| **CDR 0 vs. CDR 1** | | | | | | | | | | | | | | | |
| Age, Edu, Gender | 103.4 | 117.4 | 0.13 | 0.37 | 0.17 | 15.4 | 6 | 0.017 | 0.76 | 0.68 | 0.84 | 0.19 | 0.74 | 0.71 | 0.72 |
| + SPACE | 68.7 | 88.7 | 0.07 | 0.62 | 0.51 | 50.1 | 9 | <.001 | 0.94 | 0.91 | 0.98 | 0.11 | 1.00 | 0.85 | 0.86 |
| **CDR 0 vs. CDR 2 +** | | | | | | | | | | | | | | | |
| Age, Edu, Gender | 86.6 | 100.6 | 0.08 | 0.36 | 0.23 | 19.3 | 6 | 0.004 | 0.79 | 0.71 | 0.88 | 0.11 | 0.94 | 0.65 | 0.68 |
| + SPACE | 51.2 | 69.2 | 0.05 | 0.59 | 0.59 | 54.7 | 8 | <.001 | 0.95 | 0.91 | 0.99 | 0.09 | 0.94 | 0.88 | 0.88 |
| **CDR 0.5 vs. CDR 1** | | | | | | | | | | | | | | | |
| Age, Edu, Gender | 89.8 | 103.8 | 0.18 | 0.31 | 0.13 | 8.25 | 6 | 0.220 | 0.70 | 0.60 | 0.79 | 0.20 | 0.74 | 0.59 | 0.62 |
| + SPACE | 56.9 | 76.9 | 0.09 | 0.60 | 0.54 | 41.14 | 9 | <.001 | 0.91 | 0.86 | 0.97 | 0.11 | 0.95 | 0.73 | 0.77 |
| **CDR 0.5 vs. CDR 2 +** | | | | | | | | | | | | | | | |
| Age, Edu, Gender | 74.3 | 88.3 | 0.11 | 0.41 | 0.26 | 17.2 | 6 | 0.008 | 0.80 | 0.69 | 0.89 | 0.16 | 0.88 | 0.68 | 0.71 |
| + SPACE | 52.8 | 70.8 | 0.07 | 0.59 | 0.52 | 38.7 | 8 | <.001 | 0.91 | 0.85 | 0.98 | 0.18 | 0.88 | 0.85 | 0.86 |
| **CDR 1 vs. CDR 2 +** | | | | | | | | | | | | | | | |
| Age, Edu, Gender | 49.1 | 63.1 | 0.18 | 0.42 | 0.31 | 12.0 | 6 | 0.061 | 0.76 | 0.60 | 0.89 | 0.30 | 0.82 | 0.62 | 0.69 |
| + SPACE | 44 | 62 | 0.15 | 0.62 | 0.42 | 17.11 | 8 | 0.029 | 0.83 | 0.69 | 0.97 | 0.45 | 0.69 | 0.91 | 0.83 |

*Model predictors included (demographics only vs. demographics + SPACE); Dev deviance; AIC Akaike Information Criterion; Brier Brier score; MCC Matthews Correlation Coefficient; R²N Nagelkerke's R²; χ² model chi-square; df degrees of freedom; p significance; AUC area under the ROC curve; Upp CI / Low CI = 95% confidence interval; Opt Cutoff optimal threshold (Youden index); Sens sensitivity; Spec specificity; Acc accuracy.*

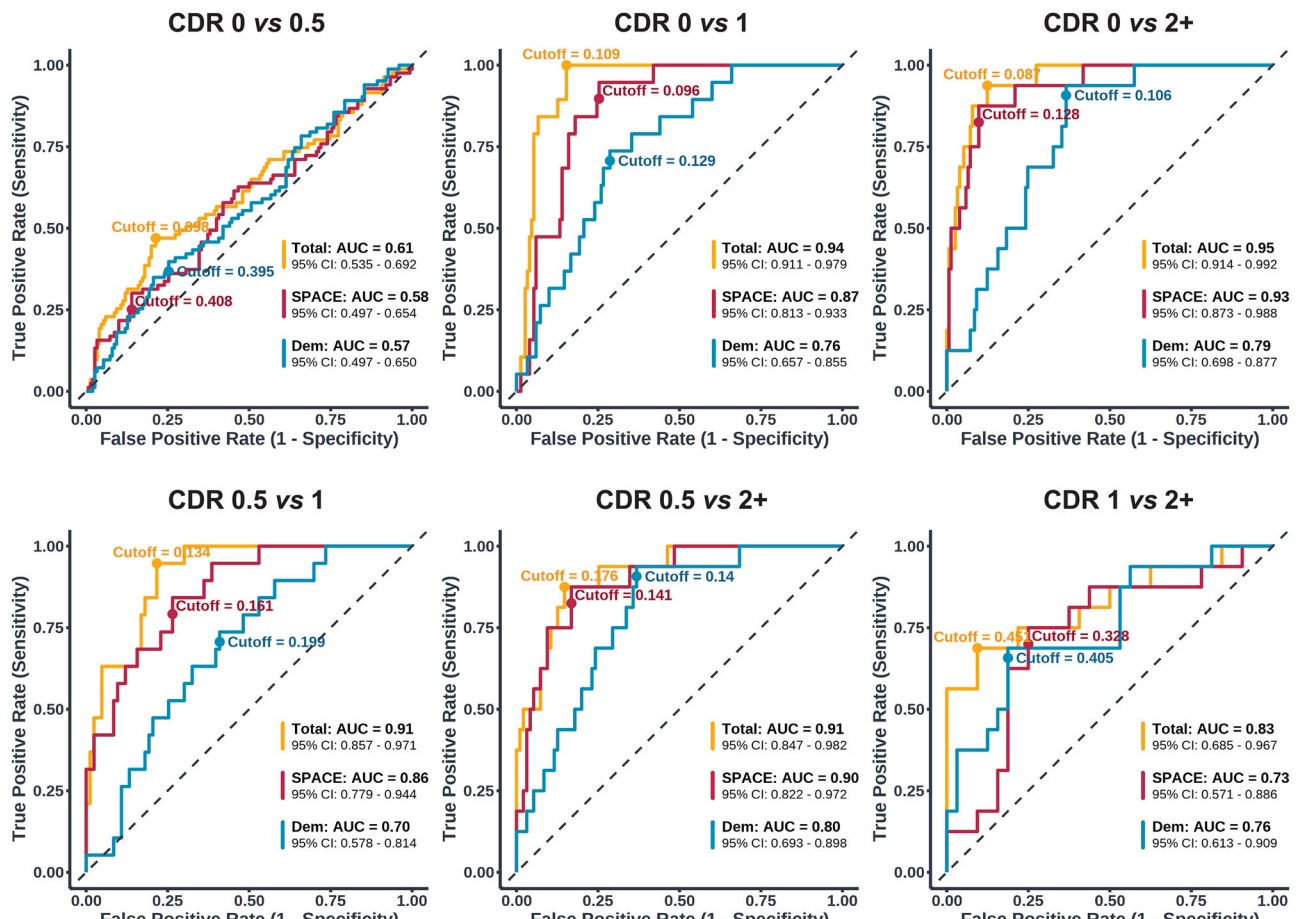

**Fig. 3 | Receiver operating Characteristic (ROC) curves comparing pairwise classification performance across contrasts between cognitive stages.** *Dem* (sociodemographic variables, blue), *SPACE* (performance on training, path integration, and perspective taking tasks, red) and *Total* (Dem + SPACE,

orange). AUC values and 95% confidence intervals (CI) are reported for each model. Cutoff values reflect optimal thresholds based on the Youden Index. The diagonal dashed lines indicate chance-level performance.

**Table 5 | Contribution of SPACE measures to model fit**

| Model | Comparison | $\Delta\chi^2$ | df | p | $p_{DeLong}$ |
|---|---|---|---|---|---|
| CDR 0 *vs.* CDR 0.5 | Age, Edu, Gender + SPACE | 4.88 | 3 | 0.181 | 0.256 |
| CDR 0 *vs.* CDR 1 | Age, Edu, Gender + SPACE | 34.7 | 3 | <.001 | <.001 |
| CDR 0 *vs.* CDR 2+ | Age, Edu, Gender + SPACE | 35.4 | 2 | <.001 | <.001 |
| CDR 0.5 *vs.* CDR 1 | Age, Edu, Gender + SPACE | 32.9 | 3 | <.001 | <.001 |
| CDR 0.5 *vs.* CDR 2+ | Age, Edu, Gender + SPACE | 21.5 | 2 | <.001 | 0.001 |
| CDR 1 *vs.* CDR 2+ | Age, Edu, Gender + SPACE | 5.07 | 2 | 0.079 | 0.138 |

Likelihood ratio $\chi^2$ tests comparing baseline demographic models to extended models including SPACE measures across clinical group contrasts.

discriminatory tasks (i.e. training time and perspective taking), sSPACE considerably reduced the overall administration time (to <11 min) of the full SPACE battery while maintaining strong diagnostic accuracy. Notably, our results revealed that sSPACE achieved AUCs of up to 0.91 in distinguishing participants with CDR 0 from those with CDR 1, and 0.82 for differentiating CDR 0.5 from CDR 1. These values compare favourably to recent digital drawing tests, such as PENSIEVE-AI, which reported an AUC of 0.93 for

distinguishing cognitively normal individuals from those with MCI/ dementia in a large community sample[66]. Critically, the PENSIEVE-AI study grouped MCI and dementia cases together and did not report classification performance for adjacent distinctions, such as between cognitively normal participants and those with MCI only. Additionally, PENSIEVE-AI still requires some degree of supervision for more severely impaired cases and relies on a stylus-enabled tablet, which may limit its scalability in non-

## Table 6 | Cross-validation performance of SPACE-based models

| Outcome | Cross Validations | | | | | | | | | |
| | K-fold | | | | | LOOCV | | | | |
| | AUC | Low CI | Upp CI | Sens | Spec | AUC | Low CI | Upp CI | Sens | Spec |
|---|---|---|---|---|---|---|---|---|---|---|
| CDR 0 *vs.* CDR 0.5 | 0.50 | 0.42 | 0.59 | 0.39 | 0.67 | 0.52 | 0.44 | 0.60 | 0.31 | 0.68 |
| CDR 0 *vs.* CDR 1 | 0.87 | 0.77 | 0.97 | 0.79 | 0.83 | 0.85 | 0.75 | 0.96 | 0.79 | 0.82 |
| CDR 0 *vs.* CDR 2+ | 0.89 | 0.81 | 0.97 | 0.69 | 0.86 | 0.88 | 0.80 | 0.96 | 0.75 | 0.87 |
| CDR 0.5 *vs.* CDR 1 | 0.80 | 0.69 | 0.92 | 0.74 | 0.75 | 0.80 | 0.68 | 0.92 | 0.63 | 0.75 |
| CDR 0.5 *vs.* CDR 2+ | 0.84 | 0.74 | 0.94 | 0.63 | 0.84 | 0.84 | 0.74 | 0.94 | 0.63 | 0.82 |
| CDR 1 *vs.* CDR 2+ | 0.71 | 0.54 | 0.88 | 0.56 | 0.88 | 0.67 | 0.49 | 0.84 | 0.50 | 0.84 |

Classification accuracy of SPACE models across clinical groups using K-fold and LOOCV, reporting AUC, sensitivity, and specificity.

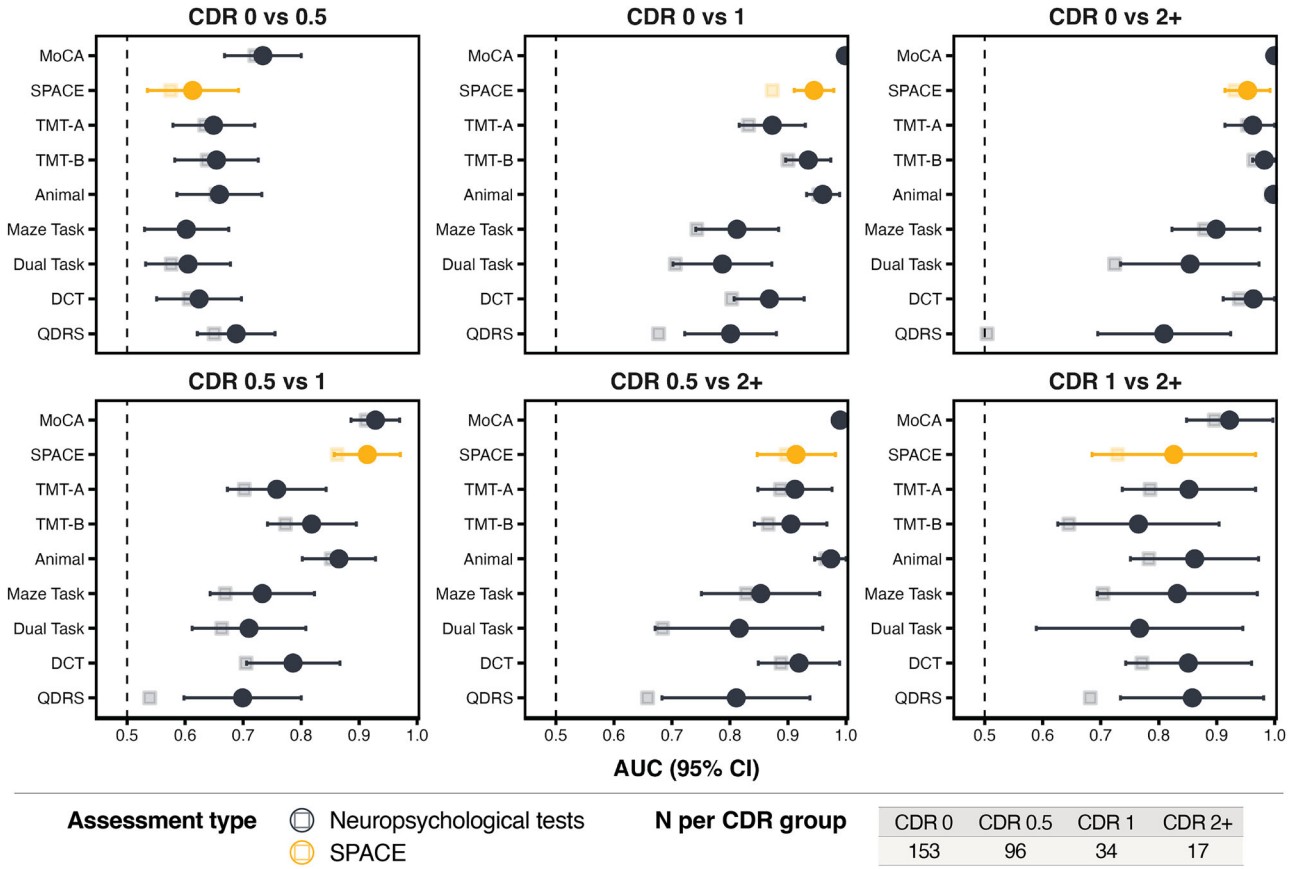

**Fig. 4 | Forest plots showing the classification performance (AUC ± 95% CI) of the SPACE tasks and neuropsychological tests across pairwise CDR group comparisons.** Each panel represents a distinct comparison. Circles denote models including demographic variables (age, gender, education), while squares denote models excluding demographic variables (test performance only). Orange circles represent SPACE, and dark blue markers represent traditional neuropsychological tests. Each point represents the estimated area under the AUC derived from logistic regression models, and horizontal lines indicate two-sided 95% confidence intervals calculated using the non-parametric DeLong method. Analyses were conducted in *n* = 300 human participants. The distribution across CDR groups is provided in the table reported in the legend of the figure.

clinical settings. In contrast, sSPACE combines strong diagnostic accuracy with the potential for fully unsupervised deployment[67], offering a practical and scalable solution for early cognitive screening. sSPACE can serve as an initial screening tool to flag individuals at risk, who may then undergo more detailed clinical assessment. However, because path integration showed greater sensitivity for discriminating between CDR 0.5 and CDR 1, administering the full SPACE battery remains advisable whenever feasible.

The tasks in SPACE were designed to engage a distributed network of brain regions along a continuum from egocentric to allocentric processing. Training engages visuospatial processing and motor control from an egocentric perspective supported by the posteromedial parietal (e.g., precuneus,

posterior cingulate) and occipital regions[68,69]. Path integration is a primarily egocentric task that relies on the updating of position and orientation based on vection (illusory sensation of self-motion), which relies on the medial entorhinal cortex. This region interacts with the hippocampus, retrosplenial and parahippocampal cortices, medial prefrontal regions, and cerebellum, in order to integrate optic-flow, vestibular, and "proprioceptive" self-motion cues to maintain spatial orientation[36,70,71]. However, path integration can also provide the self-motion framework from which allocentric representations, such as the relationships between landmarks, can emerge[72–74]. The pointing task requires participants to retrieve and transform this self-referenced information to estimate the direction of unseen landmarks,

## Table 7 | Classification performance of the short SPACE version (sSPACE)

| Outcome | No Cross Validation – sSPACE | | | Cross Validation – sSPACE | | | | | |
| --- | --- | --- | --- | --- | --- | --- | --- | --- | --- |
| | | | | K-fold | | | LOOCV | | |
| | AUC | Sens | Spec | AUC | Sens | Spec | AUC | Sens | Spec |
| CDR 0 *vs.* CDR 0.5 | 0.61 | 0.49 | 0.72 | 0.48 | 0.39 | 0.69 | 0.51 | 0.39 | 0.63 |
| CDR 0 *vs.* CDR 1 | 0.91 | 0.91 | 0.81 | 0.88 | 0.84 | 0.75 | 0.87 | 0.88 | 0.76 |
| CDR 0 *vs.* CDR 2+ | 0.95 | 0.94 | 0.88 | 0.89 | 0.69 | 0.86 | 0.88 | 0.75 | 0.87 |
| CDR 0.5 *vs.* CDR 1 | 0.82 | 0.84 | 0.78 | 0.75 | 0.59 | 0.75 | 0.75 | 0.56 | 0.75 |
| CDR 0.5 *vs.* CDR 2+ | 0.91 | 0.88 | 0.85 | 0.84 | 0.63 | 0.84 | 0.84 | 0.63 | 0.82 |
| CDR 1 *vs.* CDR 2+ | 0.83 | 0.69 | 0.91 | 0.71 | 0.56 | 0.88 | 0.67 | 0.50 | 0.84 |

Predictive performance of sSPACE models using K-fold and LOOCV procedures across clinical group comparisons.

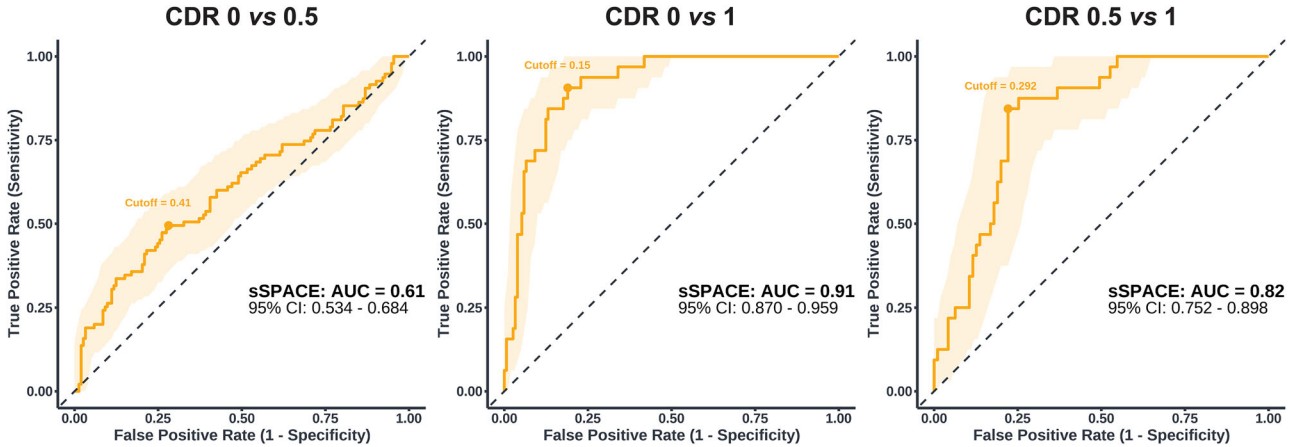

**Fig. 5 | ROC curves for the short version of SPACE (sSPACE).** ROC curves showing the classification performance of sSPACE (comprising only the training and perspective taking tasks) for three contrasts: CDR 0 *vs.* CDR 0.5, CDR 0 *vs.* CDR 1, and CDR 0.5 *vs.* CDR 1. Shaded bands indicate 95% confidence intervals around the ROC curves.

engaging both posterior parietal regions for egocentric updating (e.g., the waterfall is to my right) and retrosplenial regions for coordinate transformations (e.g., the waterfall is northeast of the rocket)[32,75]. The mapping task further demands integrating multiple viewpoints into a coherent representation of landmark locations, a process that relies on hippocampal and parahippocampal structures associated with allocentric spatial memory[76,77]. However, the mapping task can technically also be solved procedurally by reconstructing the individual paths to previously visited landmarks supported by the basal ganglia[78,79]. Finally, the perspective taking task involves mentally adopting a viewpoint other than one's own and judging where landmarks are from that imagined perspective. This task draws on allocentric representations to reconstruct spatial relations and egocentric simulation to visualise the scene from the imagined perspective. This process reflects a dynamic integration of both reference frames, engaging hippocampal and retrosplenial regions that enable flexible spatial transformations[75,80]. Together, these tasks illustrate a somewhat hierarchical but not absolute shift from self-based updating to world-based representation that underpins spatial navigation[72].

Of all the spatial navigation tasks performed in SPACE, the training phase, path integration, and perspective taking were best able to discriminate levels of cognitive impairment. These navigation impairments likely reflect early pathology in regions critical for spatial orientation and memory[16,21,31], deficits that are known to be more prominent in AD and MCI compared to normal aging[26,81]. A notable finding was the progressively longer times required to complete the SPACE training phase as the severity of cognitive impairment increased. This is likely due to deficits in both basic visuospatial integration[39,82,83] and fine motor skills for tracking objects during rotation and movement manoeuvres[69]. Similarly, a study using a

smartphone-based app called Altoida showed that a tapping task added efficacy in distinguishing between MCI and early AD[45]. More recently, research has shown that reaching movements under visual interference can distinguish MCI from healthy aging[84]. Importantly, a novel cognitive-motor framework further implicates disrupted integration of motor and spatial systems as a driver of navigation errors in early-stage AD[83]. Thus, prolonged training time in cognitively impaired participants likely reflects not only spatial learning deficits but also deterioration of visuospatial and motor skills needed to interact with the control interface.

We found that deficits in the path integration task were sensitive indicators of early signs of cognitive impairment. Path integration has been widely studied and has consistently demonstrated a strong ability to discriminate across stages of cognitive impairment[33–36]. In our cohort, individuals with mild dementia exhibited significantly greater path integration errors than those with no or questionable dementia, aligning with prior evidence[27,28]. Path integration deficits are associated with reduced grid-like representations[33,34] in the entorhinal cortex, as well as atrophy in both the entorhinal cortex[34] and hippocampus[27,28]. These deficits have been shown to emerge in individuals at genetic risk for AD, appearing well before symptom onset, even when performance on standard cognitive tests is accurate[33–35]. While these findings reinforce the interpretation of path integration tasks as sensitive markers of early AD, the inherent complexity of these tasks may limit their utility for widespread screening, particularly for older adults or unsupervised populations.

The perspective taking task in SPACE emerged as a valuable marker, with individuals in the mild and moderate dementia groups exhibiting significantly greater errors compared to those with no or questionable dementia. This task assesses the ability to mentally represent and

manipulate spatial relationships from different viewpoints, tapping into both visuospatial ability and executive function[51,77]. Indeed, research using immersive VR has found that early AD disrupts the ability to convert allocentric into egocentric information, highlighting compromised reference-frame transformations[85]. Despite its simplicity and suitability for unsupervised deployment, perspective taking remains understudied in both research and clinical screening. To our knowledge, only one study has specifically focused on perspective taking with amnestic MCI and AD participants, showing significant deficits in tasks requiring first-person viewpoint transformations[39]. In SPACE, we digitalised the Spatial Orientation Test[51], into a scalable, sensitive perspective taking assessment. However, its suitability for routine cognitive screening across diverse clinical populations requires further validation in broader clinical settings.

While the absence of significant group differences in the pointing and mapping tasks might initially appear inconsistent, these findings can be understood by considering their dependence on the path integration task. Both the pointing and mapping tasks rely on participants successfully encoding the position of the various landmarks when completing the various path integration trials. Here, participants who fail to accurately encode the locations of the landmarks are likely to perform worse on the path integration task and the following tasks (i.e. pointing and mapping) that depend on this knowledge. In contrast, performance on the perspective taking task does not depend on prior landmark encoding, as participants are provided with a map of the landmark locations. To clarify the relationship between these tasks, we conducted additional regression analyses examining whether clinical status and path integration performance jointly explained variation in pointing and mapping. These models showed that individuals who made larger errors during path integration also tended to perform worse on the pointing and mapping tasks, suggesting that difficulties in encoding landmark positions carried over into these later measures (Supplementary Note 6).

Comparing SPACE to standard neuropsychological tests allowed for direct benchmarking of its diagnostic validity. ROC analyses and DeLong tests confirmed that SPACE performed on par with, or better than, most conventional neuropsychological tests across clinical contrasts. Critically, for contrasts distinguishing CDR 0 vs 1 and 0.5 vs 1, SPACE showed comparable performance to the MoCA, TMT-B, and DCT, and outperformed the QDRS, TMT-A, Maze Test, and Dual Task. This pattern suggests that SPACE engages executive, attentional, working memory and processing speed components similar to those tapped by these tasks. The strong performance of the MoCA across all CDR stages reinforces its recognised sensitivity for detecting dementia. This strong performance was expected given the MoCA's multi-domain coverage and the fact that part of the reconstructed CDR scores were derived from clinical notes and neuropsychological tests that include the MoCA[86]. In this context, SPACE's ability to achieve AUCs close to those of the MoCA is particularly noteworthy.

Our findings build on prior work that has demonstrated that spatial navigation ability, assessed through digital tools, can be a sensitive marker of cognitive impairment[32,41,44,45,66,87–89]. Unlike traditional assessments, digital tools can enhance engagement and adherence[11,13,90,91], making them especially suitable for testing older adults in supervised and unsupervised settings. The pioneering citizen-study with Sea Hero Quest revealed the potential for scalable population-level screening, although its clinical relevance remains to be established, but see[35]. Other virtual paradigms, such as the Virtual Supermarket Test[32] and the Four Mountains Test[89], have successfully discriminated between cognitively impaired and healthy individuals[32] and shown potential to predict conversion from MCI to dementia[92]. However, these tools have been mainly confined to controlled laboratory or research contexts. SPACE aims to unify these two strands by translating the strengths of population-scale and laboratory-based approaches into a clinically validated tool, benchmarked against standard neuropsychological assessments. Here, the shorter administration times and similar diagnostic accuracy in sSPACE make it ideal for future large-scale implementation in clinics and unsupervised settings.

Despite the clear benefits of our digital assessment, several limitations should be acknowledged. First, the cross-sectional design limits conclusions about the ability of SPACE to track cognitive decline over time. Here, longitudinal studies are required to assess predictive validity. Second, although performance was robust under supervised conditions, SPACE has not yet been fully evaluated in fully unsupervised settings, where variability in digital literacy and user familiarity with touchscreen interfaces may affect task performance. To mitigate these technological barriers, we previously conducted a detailed usability study that evaluated task controls, interface aids, and the configuration of landmarks in the virtual environment across age groups[48]. This work has allowed optimisation of the current design and reduced some of these confounds. More recently, we have showed that sSPACE can reliably predict MoCA scores when deployed unsupervised[67]. Third, although the proportion of MCI cases in this study is consistent with prevalence estimates in an ageing population[93,94] and large-scale studies on digital assessment[44,66], the sample was nonetheless skewed toward individuals with CDR 0–0.5, with relatively few participants having a CDR > 1. Finally, our findings were drawn from a single study cohort. As such, replication in other samples is necessary to support generalisability as well as differences in demographic characteristics.

We acknowledge that SPACE is not intended to replace comprehensive clinical or neuropsychological assessments. While SPACE does not offer a definitive diagnosis, this work has shown its clear potential for serving as a rapid, scalable tool for widespread screening of cognitive impairment. To maximise clinical relevance, SPACE and sSPACE metrics could also be integrated with complementary data sources such as neuropsychological assessments, fluid and imaging biomarkers, genetic profiles, and other clinical data[95]. A promising future direction is the continuous monitoring of spatial navigation ability, supported by mobile sensor technologies and integrated health data[47,96,97]. This approach could enable early detection of subtle cognitive decline, support longitudinal tracking of neurological changes, and contribute to digital phenotyping for MCI. Incorporating these tools into clinical workflows and public health systems may ultimately strengthen early detection strategies and support timely, targeted interventions.

In summary, this study provides clear evidence that spatial navigation tasks are a sensitive marker for detecting early cognitive impairment. Through immersive and engaging tasks that tap a distributed network of brain regions associated with spatial and navigation abilities, the tablet-based SPACE tool provides a domain-specific and cost-effective alternative to traditional clinical assessments. Our results demonstrate strong diagnostic accuracy across cognitive levels, particularly across the questionable and mild levels of impairment. The shortened version, sSPACE, maintains performance while reducing assessment time, supporting its suitability for widespread and unsupervised deployment. This approach can alleviate pressure on healthcare systems and extend reach to underserved populations. While additional validation in unsupervised environments is clearly needed, SPACE offers promising avenues for expanding access to cognitive screening and advancing early intervention strategies.

## Data availability

The data used in this study were collected under ethical approval from the NHG DSRB, Singapore (reference number: 2021/01160). Due to participant confidentiality and institutional data protection policies, the data are not publicly available. De-identified data may be made available to qualified researchers for research purposes upon reasonable request, subject to approval by the NHG DSRB and execution of an appropriate institutional data use agreement. Access may be restricted to non-commercial research use and may require compliance with local data protection regulations. Requests for access should be directed to the corresponding authors: Giorgio Colombo: gicolombo@ethz.ch Victor R. Schinazi: vschinaz@bond.edu.au. The authors will acknowledge receipt of requests within two weeks and aim to provide a decision regarding data access within four weeks, contingent

upon DSRB review and institutional requirements. Source data underlying Figs. 2 and 4 are available in the Figshare repository at https://doi.org/10.6084/m9.figshare.31119679[98].

## Code availability

The code used for the statistical analyses reported in this manuscript is publicly available in the Figshare repository at https://doi.org/10.6084/m9.figshare.31119679[98]. The analyses were implemented in R using publicly available packages and executed in RStudio (version 2024.12.0 + 467).

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

## Acknowledgements

This research is supported by the National Research Foundation Singapore (NRF) under its Campus for Research Excellence and Technological Enterprise (CREATE) programme.

## Author contributions

G.C., K.M., and V.R.S. conceived the study. G.C., J.G., and V.R.S. were responsible for software conceptualisation and development. E.C., J.R.C., and M.K.P.L. were responsible for participant recruitment and data collection. E.C., C.P.C., M.J.H.L., and P.N.G.G. provided clinical oversight and contributed to data collection coordination. G.C. and V.R.S. curated and analysed the data. G.C., K.M., and V.R.S. contributed to data interpretation. K.M. procured ethics approval. G.C. wrote the first draft of the manuscript and prepared the data visualisations. G.C., W.R.T., and V.R.S. contributed to critical revisions and substantive refinement of the manuscript. V.R.S. supervised the project. All authors reviewed and approved the final manuscript.

## Funding

## Competing interests

The authors declare no competing interests.
