## [Transparent Peer Review file · Communications Medicine]

Spatial Navigation as a Digital Marker for Clinically Differentiating Cognitive Impairment Severity

Corresponding Author: Dr Giorgio Colombo

Version 0:

Reviewer comments:

Reviewer #1

(Remarks to the Author)

Major findings that SPACE a digital tool to evaluate spatial navigation was successful in categorising dementia severity in line with CDR classification. SPACE significantly increased the AUC for distinguishing normal cognition from mild impairment, normal cognition, moderate impairment, and very-mild from mild, all with high sensitivity and specificity. A short version of SPACE (< 11 mins) reduced administration time, showing greater clinical utility while maintaining diagnostic accuracy.

Very interesting study, well written and well reported.

To improve this manuscript please consider

1. Providing a table of demographic characteristics by dementia severity according to CDR (age, sex, dementia duration are important factors in this)
2. Providing greater clarity of how you controlled for age and sex related performance differences - particularly important when measuring spatial navigation processes – see Coutrot, A., Silva, R., Manley, E., de Cothi, W., Sami, S., Bohbot, V. D., et al. (2018). Global determinants of navigation ability. *Curr. Biol.* 28, 2861–2866. doi: 10.1016/j.cub.2018.06.009
3. Have you considered using ANCOVA with age and sex as covariates when exploring group differences?
4. Providing greater clarity on how patients were diagnosed in the memory clinic. Are all patients AD/ MCI or is there mixed pathology here too? Important because different pathology can present differing spatial deficits see, Lowry, E., Puthusseryppady, V., Coughlan, G., Jeffs, S., & Hornberger, M. (2020). Path integration changes as a cognitive marker for vascular cognitive impairment?—A pilot study. *Frontiers in Human Neuroscience*, 14, 131.
5. A battery of psych assessments were used and APOE4 collected were these considered in this study? If so, what were the outcomes? Did you explore TMT, maze task and others with CDR categorisation and SPACE performance? Processing speed and executive function key components to findings re training time/increased processing speed
6. The paper would benefit from more discussion about significant domains identified of SPACE sig group differences were found in training time ($F(3, 75, 296) = 14.7, p < 0.001, \xi = 0.657$), path integration distance error ($F(3, 296) = 7.0, p = 0.004, \xi = 0.587$), and 76 perspective taking error ($F(3, 296) = 11.0, p < 0.001, \xi = 0.51$). No significant group differences were 77 observed for egocentric pointing error ($F(3, 296) = 2.61, p = 0.101$) or mapping accuracy - how do you interpret this? Further clarity on these domains and associated neuroanatomy/ potential deficits – is this a strictly allocentric/ path integration deficit? Does perspective taking tap into egocentric processing also?
7. Task appears very similar to domains tapped in spatial app Sea Hero Quest, consider how this literature/ and others may inform/ complement/ make findings novel - G. Coughlan, A. Coutrot, M. Khondoker, A. Minihane, H. Spiers, & M. Hornberger, Toward personalized cognitive diagnostics of at-genetic-risk Alzheimer's disease, *Proc. Natl. Acad. Sci. U.S.A.* 116 (19) 9285-9292, <https://doi.org/10.1073/pnas.1901600116> (2019).

Reviewer #2

(Remarks to the Author)

Thank you for letting me review this interesting article investigating a new cognitive task for detecting cognitive impairment in dementia. Overall, this is a well conducted study in a large sample size, showing good discrimination of cognitive impairment and healthy. However, there are a few issues with the study which might warrant further revision.

- Overall, the clinical characterisation of the cohort is poor, considering that this is meant to detect cognitive impairment in dementia. There is no information provided on which clinical diagnoses people had. Also, the authors should provide the diagnostic criteria which were applied to the sample and whether the diagnoses were biomarker confirmed (it seems this

data was collected but not presented).

- Along the same lines, over 80% of the sample has a CDR score of 0 or 0.5, which is considered to be healthy or having only very minor cognitive changes, not qualifying for a diagnosis of dementia. This large discrepancy between a large healthy group and people who have cognitive impairment (CDR >1) clearly questions in how far the cognitive impairment group was representative.
- It is not clear which cognitive profile/deficits the cohort has, since there is no other cognitive data provided. There should be at least a cognitive screening tests, e.g. MoCA, or even better the cohort should have been characterised by neuropsychological testing, to determine which cognitive deficits they had.
- Such other cognitive testing would have also allowed how the SPACE battery compares to existing gold standard cognitive testing, which is in the current version not clear. The cognitive impairment is solely based on the CDR, which is a clinical assessment and not a cognitive assessment per se.
- The authors show that the path integration condition is one of the most sensitive to detect cognitive impairment but then remove this condition from the short version of the test. This seems an odd decision, solely based on time constraints for testing. Would it not make sense to keep the most sensitive measures, regardless of time taken for testing? Please provide a more detailed rationale.
- When comparing AUC or model performance for mild and moderate, or normal and very mild, there is some inconsistency in the wording that sometimes suggest it is significant while other times it is not. Please check this carefully throughout the manuscript.
- The paragraph explaining SPACE classification performance for normal vs very-mild impairment needs to be incorporated earlier for better context moving forward. Also, the explanation given as to why the results might not be significant is insufficient. It would be better if the authors could find an explanation within the SPACE tasks themselves, instead of a very vague explanation.
- Finally, it is odd to state that "SPACE consistently improved model performance compared to sociodemographic variables alone." Usually, no classification would be only conducted on sociodemographic variables, since diagnostic, cognitive and functional variables usually determine this.

Reviewer #3

(Remarks to the Author)

I co-reviewed this manuscript with one of the reviewers who provided the listed reports. This is part of the Communications Medicine initiative to facilitate training in peer review and to provide appropriate recognition for Early Career Researchers who co-review manuscripts.

Reviewer #4

(Remarks to the Author)

In this large clinimetric study, the authors evaluated the discriminatory power of the Spatial Performance Assessment for Cognitive Evaluation (SPACE), a novel and promising tablet-based tool for assessing spatial navigation. The tool appears impressive in distinguishing individuals with varying degrees of cognitive impairment. SPACE was tested in both a memory clinic and a community cohort, including 300 participants. I have carefully reviewed the manuscript. Below are my impressions and my suggestions.

Major concerns

From a clinimetric standpoint, the tablet-based paradigm devised by the authors shows impressive performance in terms of classification accuracy. Nevertheless, after careful examination of the manuscript, several concerns arise:

-First, it is puzzling that, despite the availability of extensive neuropsychological and neurobiological data, patient classification remains rudimentary, with no reference to clinical phenotypes or etiological diagnoses. Patient classification indeed relies only on the CDR score.

-Second, why was the "SPACE" model not compared to a control/reference model incorporating standard cognitive measures, such as the MoCA?

-Third, critical variables—such as tablet experience, depression, anxiety, and sleep—are neither reported nor screened for their eligibility as covariates within the authors' proposed models.

-Similarly, data on fluid biomarkers and APOE status were not included in the analysis. Why?

These are serious flaws that constitute the principal limitations of this study. Without supplementary analyses incorporating these variables or a well-reasoned justification for their exclusion, the scientific validity of the work is weak.

Minor concerns

Introduction

-In lines 28–32, the authors claim that multidomain cognitive impairment may become detectable, especially when the neurodegenerative disease reaches an advanced stage. This statement is not necessarily true and should be more carefully contextualized. Clinically, the primary issue lies in the limited sensitivity of standard neuropsychological assessments, which

often fail to capture early or subtle cognitive deficits. Furthermore, references 11 and 12 concern spatial navigation and do not support the authors' statement. I suggest consulting the following references for a more theoretical discussion of this crucial point.

Suggested references:

Ilardi et al. J Alzheimers Dis. doi:10.3233/JAD-240339

Methods

-It is somewhat surprising that the study protocol includes a gait assessment but does not incorporate a comprehensive evaluation of upper extremity motor function.

Results: group differences in spatial navigation

-Why were group differences analyzed using a one-way ANOVA with bootstrapping? Is there a specific rationale behind this methodological choice?

-How did the authors compute the effect size (ξ)? Please, describe how this measure was quantified, as well as any conventional rules of thumb for interpreting its magnitude.

-Table 2 is informative, but it is likely more appropriate to be placed in the supplementary materials. I recommend that the authors concisely summarize these results in the main text and present the between-group comparisons in a graphical format.

-Which method was used to conduct the post hoc tests? This relevant information is not reported.

-The post hoc comparisons are reported using p-values as if they indicated effect size. The authors should report effect sizes for each pairwise comparison and restructure this results section by including a qualitative interpretation of the effects' magnitude (e.g., small, moderate, large).

-Line 93-103: Redundant. The same information is already provided in the Statistical Analyses section.

Discussion

As compared to the well-written introduction, the discussion is verbose and lacks breadth. In my opinion, the authors should emphasize the potential for such a serious game to increase patients' adherence to the assessment setting. Additionally, both the introduction and discussion lack theoretical references to egocentric and allocentric reference frames, which are specifically probed by the pointing task and the perspective-taking task, respectively. For reference purposes, please consult:

Cavanna & Trimble. Brain. 2006. doi:10.1093/brain/awl004

Ilardi et al. Aging Clin Exp Res. 2022. doi:10.1007/s40520-021-01930-y

Ruggiero et al. Behav Brain Res. 2020. doi:10.1016/j.bbr.2020.112793

Wood et al. Front Neurol. 2016. doi:10.3389/fneur.2016.00215

I would like to thank the Editor for inviting me to review this interesting manuscript. The authors must address and justify the major methodological shortcomings in an otherwise promising paper.

Sincerely,

Dr. Ciro Rosario Ilardi

Version 1:

Reviewer comments:

Reviewer #1

(Remarks to the Author)

I am satisfied with the authors responses. Thank you for taking on feedback and providing such comprehensive additional analysis.

Reviewer #2

(Remarks to the Author)

The authors have addressed all my comments sufficiently.

Reviewer #4

(Remarks to the Author)

The authors have responded to all of my comments with clarity, professionalism, and a constructive spirit. The revisions are satisfactory and align well with the high standards expected of a Nature Portfolio journal.

Although brief paper-and-pencil cognitive screeners such as the MoCA often show discrimination power comparable to more complex digital tasks like SPACE (a finding increasingly supported by the recent reference literature, to which this study contributes), the paradigm proposed by the authors is compelling; it may enhance patient engagement and represents a meaningful innovation in the field of digital neuropsychology applied to dementia diagnostics.

I commend the authors for their work and wish them the best as this line of research continues to develop.

Reply to reviewers

Referee expertise: Referee #1: Cognitive decline scoring, spatial navigation; Referee #2: Dementia, navigation defects; Referee #3: Early career researcher; Referee #4: Cognitive impairment

Reviewer #1:

Major findings that SPACE, a digital tool to evaluate spatial navigation, was successful in categorising dementia severity in line with CDR classification. SPACE significantly increased the AUC for distinguishing normal cognition from mild impairment, normal cognition, moderate impairment, and very-mild from mild, all with high sensitivity and specificity. A short version of SPACE (< 11 mins) reduced administration time, showing greater clinical utility while maintaining diagnostic accuracy. Very interesting study, well written and well reported.

To improve this manuscript, please consider:

- 1. Providing a table of demographic characteristics by dementia severity according to CDR (age, sex, and dementia duration are important factors in this).**

We thank the reviewer for this constructive comment and agree that adding these details is important. In the revised manuscript, we now provide a demographic table stratified by CDR severity, and report age, sex, and education for each CDR level. To give fuller context and satisfy several comments that we received on the poor clinical characterisation of our sample, Table 1 now also includes clinical diagnosis, aetiology, significant cerebrovascular disease (CeVD) status, p-tau217 risk category, APOE genotype, standard neuropsychological measures (MoCA, QDRS, Trail Making A/B, Animal Fluency, Digit Cancellation, Dual-task, Maze task), and the SPACE tasks. This table provides a clearer clinical and cognitive characterisation of our sample (see Table 1).

Table 1 | Sample characteristics. Demographic, clinical, neuropsychological, and task-performance characterisation of the sample by CDR status.

Characterisation	Variable	Clinical Dementia Rating (CDR)					Overall, N = 300	p-value
		CDR=0, N = 153 (51%)	CDR=0.5, N = 96 (32%)	CDR=1, N = 34 (11%)	CDR=2, N = 14 (5%)	CDR=3, N = 3 (1%)		
Demographic	Age, years	74 (69, 78)	75 (70, 78)	75 (69, 78)	78 (73, 79)	87 (81, 88)	74 (70, 78)	0.079
	Gender, n (%)							0.4
	Male	63 (41%)	40 (42%)	17 (50%)	4 (29%)	0 (0%)	124 (41%)	
	Female	90 (59%)	56 (58%)	17 (50%)	10 (71%)	3 (100%)	176 (59%)	
	Education level, n (%)							0.009
	No formal education	6 (3.9%)	5 (5.2%)	0 (0%)	2 (14%)	0 (0%)	13 (4.3%)	
	Primary school	13 (8.5%)	12 (13%)	10 (29%)	3 (21%)	1 (33%)	39 (13%)	
	Secondary school	45 (29%)	32 (33%)	15 (44%)	7 (50%)	1 (33%)	100 (33%)	
	High school	35 (23%)	24 (25%)	6 (18%)	2 (14%)	1 (33%)	68 (23%)	
	University	54 (35%)	23 (24%)	3 (8.8%)	0 (0%)	0 (0%)	80 (27%)	
	Depression, score	1.00 (1.00, 2.00)	1.00 (1.00, 4.00)	1.00 (1.00, 1.75)	1.00 (1.00, 1.75)	1.00 (1.00, 3.00)	1.00 (1.00, 2.13)	0.10
	Anxiety, score	2.00 (1.00, 3.00)	2.00 (1.00, 3.63)	1.00 (1.00, 2.00)	1.00 (1.00, 2.00)	1.00 (1.00, 2.50)	1.00 (1.00, 3.00)	0.14
	Stress, score	1.00 (1.00, 3.00)	1.00 (1.00, 3.25)	1.00 (1.00, 2.00)	1.00 (1.00, 2.75)	1.00 (1.00, 2.00)	1.00 (1.00, 3.00)	0.6
	Sleep, hours	6.50 (5.50, 7.50)	7.00 (5.50, 8.00)	8.00 (7.00, 9.75)	9.25 (8.00, 10.00)	9.00 (8.50, 10.50)	7.00 (6.00, 8.00)	<0.001
	Tablet experience, n (%)							<0.001
None	12 (7.8%)	24 (25%)	13 (38%)	10 (71%)	2 (67%)	61 (20%)		
Low	48 (31%)	31 (32%)	17 (50%)	4 (29%)	1 (33%)	101 (34%)		
High	93 (61%)	41 (43%)	4 (12%)	0 (0%)	0 (0%)	138 (46%)		
SBSOD, score	3.57 (2.79, 4.36)	3.93 (3.13, 4.64)	3.75 (3.16, 4.21)	4.46 (3.59, 5.48)	6.86 (6.14, 6.93)	3.71 (2.93, 4.50)	0.001	
Clinical	Clinical diagnosis, n (%)							<0.001
	NCI	86 (56%)	21 (22%)	0 (0%)	0 (0%)	0 (0%)	107 (36%)	
	SCD/SCI	50 (33%)	17 (18%)	0 (0%)	0 (0%)	0 (0%)	67 (22%)	
	MCI	17 (11%)	54 (56%)	2 (5.9%)	0 (0%)	0 (0%)	73 (24%)	
	Dementia	0 (0%)	4 (4.2%)	32 (94%)	14 (100%)	3 (100%)	53 (18%)	
	Aetiology, n (%)							<0.001
	AD, Non-Vascular	17 (11%)	15 (16%)	18 (53%)	5 (36%)	1 (33%)	56 (19%)	
	AD, Vascular	3 (2.0%)	7 (7.3%)	5 (15%)	8 (57%)	2 (67%)	25 (8.3%)	
	Non-AD, Non-Vascular	103 (67%)	52 (54%)	5 (15%)	0 (0%)	0 (0%)	160 (53%)	
	Non-AD, Vascular	30 (20%)	22 (23%)	6 (18%)	1 (7.1%)	0 (0%)	59 (20%)	
	Significant CeVD status, n (%)							0.003
	No	120 (78%)	67 (70%)	23 (68%)	5 (36%)	1 (33%)	216 (72%)	
	Yes	33 (22%)	29 (30%)	11 (32%)	9 (64%)	2 (67%)	84 (28%)	
	pTau risk, n (%)							<0.001
	High	27 (18%)	26 (27%)	25 (74%)	14 (100%)	3 (100%)	95 (32%)	
Low	126 (82%)	70 (73%)	9 (26%)	0 (0%)	0 (0%)	205 (68%)		
Genotype, n							0.2	
E2/E2	3 (6.1%)	0 (0%)	0 (0%)	0 (0%)	0 (0%)	3 (3.1%)		
E2/E3	5 (10%)	2 (6.7%)	1 (10%)	1 (17%)	0 (0%)	9 (9.4%)		

		Clinical Dementia Rating (CDR)						
Characterisation	Variable	CDR=0, N = 153 (51%)	CDR=0.5, N = 96 (32%)	CDR=1, N = 34 (11%)	CDR=2, N = 14 (5%)	CDR=3, N = 3 (1%)	Overall, N = 300	p-value
	E3/E3	32 (65%)	17 (57%)	4 (40%)	4 (67%)	0 (0%)	57 (59%)	
	E3/E4	9 (18%)	10 (33%)	5 (50%)	0 (0%)	1 (100%)	25 (26%)	
	E4/E4	0 (0%)	1 (3.3%)	0 (0%)	1 (17%)	0 (0%)	2 (2.1%)	
Neuropsychological	MoCA, score	27.0 (26.0, 28.0)	25.5 (20.8, 27.0)	16.0 (12.3, 17.8)	9.5 (7.3, 12.0)	5.0 (3.0, 5.5)	26.0 (20.0, 28.0)	<0.001
	MoCA (education adj.), score	28.0 (27.0, 29.0)	26.0 (20.8, 27.0)	16.0 (13.3, 18.0)	10.5 (8.3, 12.8)	6.0 (4.0, 6.0)	27.0 (21.0, 28.0)	<0.001
	QDRS informant, score	0.50 (0.00, 1.50)	1.50 (0.50, 3.00)	2.00 (0.50, 3.50)	0.75 (0.00, 1.13)	2.00 (1.00, 3.00)	1.00 (0.00, 2.25)	<0.001
	Trail Making A, seconds	35 (27, 47)	44 (31, 64)	70 (49, 90)	150 (75, 150)	150 (150, 150)	41 (29, 66)	<0.001
	Trail Making B, seconds	89 (68, 136)	117 (78, 261)	300 (156, 300)	300 (300, 300)	300 (300, 300)	106 (75, 240)	<0.001
	Digit Cancellation Test, n	24 (20, 29)	21 (17, 28)	17 (11, 20)	9 (6, 12)	3 (2, 12)	22 (17, 28)	<0.001
	Animal Fluency, n	17 (15, 21)	15 (11, 19)	9 (6, 11)	6 (3, 8)	0 (0, 0)	15 (11, 19)	<0.001
	Dual task, n	94 (86, 102)	92 (85, 98)	81 (72, 95)	88 (70, 94)	44 (22, 67)	92 (84, 100)	<0.001
	Maze task, seconds	25 (18, 35)	29 (22, 40)	49 (27, 60)	63 (33, 240)	240 (240, 240)	28 (21, 44)	<0.001
SPACE	Training, seconds	262 (238, 293)	274 (245, 313)	309 (282, 364)	352 (318, 404)	424 (376, 436)	274 (246, 312)	<0.001
	Path integration, meters	196 (157, 261)	212 (170, 240)	290 (236, 372)	308 (236, 379)	–	205 (161, 273)	<0.001
	Pointing, degrees	76 (65, 89)	78 (67, 94)	92 (74, 102)	102 (98, 106)	–	79 (67, 92)	0.016
	Mapping, R ²	0.56 (0.30, 0.86)	0.57 (0.24, 0.84)	0.77 (0.40, 0.85)	0.80 (0.80, 0.81)	–	0.57 (0.27, 0.85)	0.5
	Memory, % correct							<0.001
	0	1 (0.7%)	1 (1.2%)	4 (20%)	0 (0%)	–	6 (2.4%)	
	33	7 (4.7%)	8 (9.6%)	3 (15%)	1 (50%)	–	19 (7.5%)	
	100	142 (95%)	74 (89%)	13 (65%)	1 (50%)	–	230 (90%)	
	Perspective taking, degrees	42 (28, 58)	43 (31, 57)	58 (49, 67)	60 (54, 72)	58 (51, 66)	46 (32, 61)	<0.001

Note. For continuous variables we report the median (IQR). For categorical variables we report the raw number (%). The p-value was calculated using the Kruskal-Wallis rank sum test or Pearson's Chi-squared test. Supplementary Table 1 details duration of dementia.

Regarding dementia duration, we obtained the year of each participant’s first clinical diagnosis and calculated duration (in full years) for all individuals with dementia. Because the year of diagnosis may lag behind symptom onset and documentation practices differ across aetiologies and sites, this measure represents an imperfect proxy for true disease duration. To avoid overstating its precision, we did not include duration in Table 1. Instead, we report these data in the Supplementary Information (Supplementary Table 1). The distribution of years since diagnosis across the dementia group was as follows: 0–1 year (n = 17), 2–4 years (n = 28), 5–9 years (n = 6), and ≥ 10 years (n = 2). These values should be interpreted with care since they reflect the time since the documented diagnosis, rather than the actual disease duration.

Supplementary Table 1 | Distribution of dementia duration in the clinical sample. Years since diagnosis of dementia among participants classified with dementia, presented by duration category.

Dementia duration	n	mean	median	SD
0–1 year	17	0.8	1.0	0.4
2–4 years	28	2.8	2.0	0.9
5–9 years	6	7.5	8.0	1.6
≥10 years	2	13.5	13.5	0.7
Overall	53	3.1	2.0	3.0

2. Providing greater clarity of how you controlled for age and sex related performance differences - particularly important when measuring spatial navigation processes – see Coutrot, A., Silva, R., Manley, E., de Cothi, W., Sami, S., Bohbot, V. D., et al. (2018). Global determinants of navigation ability. *Curr. Biol.* 28, 2861–2866. doi: 10.1016/j.cub.2018.06.009. Have you considered using ANCOVA with age and sex as covariates when exploring group differences?

We agree with the reviewer about the importance of controlling for demographic variables when analysing spatial navigation performance. As detailed in the Supplementary Information (Supplementary Table 3), we already included age and gender as covariates in all logistic regression models predicting clinical classification. Regression coefficients, standard errors, p-values, and odds ratios for these variables are reported for every CDR contrast. Moreover, we benchmarked classification accuracy using three complementary models: a Demographic model, including age, gender, education (*Dem*), a model including only SPACE tasks (*SPACE*), and a combined model including both demographics and the tasks in SPACE (*Total*). This approach ensures that the diagnostic performance of SPACE reflects variance attributable to spatial navigation beyond demographic effects.

That said, the reviewer is correct in pointing out that controlling for these variables should be done already in the feature selection stage, prior to running AUC models. We have now conducted an ANCOVA for each SPACE task, including age and gender as covariates, to estimate the main effect of clinical severity (CDR) while controlling for demographic variability. Because gender variability was limited (F = 13, M = 4) among participants with a CDR of 2+, this group was excluded from the ANCOVA comparisons, as the models could not converge.

Following standard recommendations, we first verified the assumption of homogeneity of regression slopes by testing the *age* × *CDR* and *age* × *gender* interactions. These interactions were not significant for all tasks (all *ps* > .10) and were removed to maintain model parsimony, except for the memory task, where the *age* × *CDR* interaction was significant ($F_{2, 243} = 12.41, p < .001$). Supplementary Table 8 summarises the results of the ANCOVAs. After accounting for age and gender, the main effect of CDR remained significant for the training, path integration and perspective taking tasks (all *ps* < .001). Additionally, CDR was significant for the pointing (*p* = .021) and memory tasks (*p* < .001). However, as described in the manuscript, the memory task was excluded from the main analyses because of the imbalance and ceiling effect of this variable.

While we acknowledge the importance of running the ANCOVAs and controlling for age and gender, we believe that retaining the robust ANOVAs as our primary analytical method is more appropriate. The robust ANOVA provides a more reliable and unbiased estimate when sample sizes vary across groups and when the assumptions of normality and homogeneity of variance are violated, which was the case for most models. However, as the ANCOVA suggests the inclusion of the pointing task, we further conducted ROC analyses with this task and report them in the Supplementary Information (Supplementary Table 9). Here, we note that the inclusion of the pointing task does not alter our classification accuracy.

Supplementary Table 8 | Results of ANCOVAs examining the main effect of clinical severity (CDR) on performance across SPACE tasks while controlling for Age and Gender. For each task, the table reports the Sum of Squares, degrees of freedom (df), Mean Square, F values, and corresponding p values.

Task	Effect	Sum of Squares	df	Mean Square	F	p
Training	CDR	83,225	2	41,612	17.904	< .001
	Age	22,471	1	22,471	9.668	.002
	Gender	2,222	1	2,222	0.956	.329
	CDR * Gender	4,153	2	2,077	0.893	.410
	Residuals	641,492	276	2,324		
PI Distance	CDR	150,912	2	75,456	10.693	< .001
	Age	190,635	1	190,635	27.017	< .001
	Gender	7,588	1	7,588	1.075	.301
	CDR * Gender	3,052	2	1,526	0.216	.806
	Residuals	1,750,000	248	7,056		
Pointing	CDR	2,664.00	2	1,332.00	3.934	.021
	Age	136.89	1	136.89	0.404	.525
	Gender	1.74	1	1.74	0.005	.943
	CDR * Gender	430.32	2	215.16	0.635	.531
	Residuals	83,641.38	247	338.63		
Map	CDR	0.2145	2	0.1072	1.175	.310
	Age	0.0866	1	0.0866	0.949	.331
	Gender	0.0123	1	0.0123	0.135	.714
	CDR * Gender	0.5881	2	0.2941	3.223	.042
	Residuals	22.4473	246	0.0912		
Memory Correct	CDR	12,270.0	2	6,135.0	13.351	< .001
	Age	85.8	1	85.8	0.187	.666
	Gender	158.0	1	158.0	0.344	.558
	CDR * Gender	1,783.4	2	891.7	1.941	.146
	Residuals	113,042.0	246	459.5		
Perspective taking	CDR	5,636	2	2,818.2	7.489	< .001
	Age	4,955	1	4,954.5	13.166	< .001
	Gender	401	1	401.3	1.066	.303
	CDR * Gender	101	2	50.5	0.134	.875
	Residuals	102,733	273	376.3		

Supplementary Table 9 | ROC analysis for models incorporating SPACE including the pointing task. This analysis excludes CDR contrast of CDR 2+ due to limited number of participants playing the pointing task.

Outcome	AUC	CI low	CI high	Cutoff	Sens	Spec	N
CDR 0 vs 0.5	0.597	0.520	0.675	0.472	0.232	0.953	232
CDR 0 vs 1	0.942	0.904	0.979	0.117	0.941	0.860	167
CDR 0.5 vs 1	0.909	0.850	0.968	0.112	1.000	0.744	99

We have edited the Methods section of the manuscript to reflect these changes:

Methods: Statistical analyses

All statistical analyses and plots were performed using Jamovi (v.2.6.26.0) and RStudio (v.2024.12.0+467), with p-values < 0.05 considered statistically significant, and Bonferroni corrections were applied whenever appropriate. Descriptive statistics were computed for key demographic, clinical, neuropsychological, and task-

performance variables across CDR levels. Given violations of the assumptions of normality of residuals and homogeneity of variance, we examined group differences in performance on the spatial navigation tasks of SPACE using robust one-way ANOVAs (modified one-step estimator; 5,000 bootstrap samples). To account for age and gender differences in navigation performance, we additionally conducted ANCOVAs for each SPACE task with these variables included as covariates. As the ANCOVAs supported the group differences reported with the robust ANOVAs, we retained the robust ANOVA as our primary approach. Robust ANOVAs provide unbiased estimates under violations of normality, homogeneity of variance, and unequal group sizes (Field & Wilcox, 2017), which was the case for most of our models. We report the results of the ANCOVA in the Supplementary Information G for transparency and comparison.

Logistic regressions were then used to model the relationship between predictors and all combinations of binary outcomes from the CDR scale. Following the methodology from previous researchers (Berron et al., 2024), we compared three model types: a demographic model with age, education and gender (Dem), a model including only the significant tasks in SPACE (SPACE), and a combined model including both demographic variables and the SPACE tasks (Total). The purpose of this comparison was to establish a baseline model using known demographic risk factors for dementia, against which the added discriminative value of SPACE performance could be evaluated. Model performance was compared using DeLong's test to evaluate the incremental contribution of SPACE beyond demographic predictors. The optimal classification cutoff for each model was determined using the Youden index and assessed using the AUC, sensitivity, specificity, and accuracy. Model performance was further evaluated using deviance, AIC, and the Brier score. Given that only two participants with a CDR of 2+ completed the path integration, pointing, and mapping tasks, this group were excluded from the analysis for these tasks. To account for the results of the ANCOVA, we additionally evaluated a SPACE model that included the pointing task alongside demographic variables (Supplementary Table 9).

- 4. Providing greater clarity on how patients were diagnosed in the memory clinic. Are all patients AD/MCI, or is there mixed pathology here too? Important because different pathology can present differing spatial deficits see, Lowry, E., Puthusserypady, V., Coughlan, G., Jeffs, S., & Hornberger, M. (2020). Path integration changes as a cognitive marker for vascular cognitive impairment?—A pilot study. *Frontiers in Human Neuroscience*, 14, 131.**

We thank the reviewer for this helpful comment and have expanded the description of the diagnostic procedures in the revised manuscript for greater clarity. The patient sample was recruited from the National University Hospital (NUH) and St. Luke's Hospital Memory Clinics in Singapore. Control participants were recruited from these two sites and the community. All diagnoses were established by clinical consensus at the Memory, Aging & Cognition Centre (MACC), involving neurologists and research staff who reviewed clinical notes, neuropsychological assessments, and neuroimaging and blood biomarkers when available. Following reviewers' comments, we now report CDR scores and diagnosis from clinical consensus.

The CDR was derived following standard scoring procedures (Morris, 1993) and was either taken from existing study assessments within one year of the SPACE visit or reconstructed from comprehensive clinical documentation and neuropsychological data when unavailable. This approach has been shown to achieve good-to-excellent agreement (ICC = 0.81–0.92) with face-to-face assessments (Dauphinot et al., 2024).

The diagnostic consensus classified participants into groups reflecting cognitive severity (NCI = No Cognitive Impairment, SCD/SCI = Subjective Cognitive Decline/Impairment, MCI = Mild Cognitive Impairment, and Dementia) and aetiology (AD, vascular; non-AD/non-vascular; mixed AD + vascular), following the recommendation of the National Institute on Aging and the Alzheimer's Association (NIA-AA) for the diagnosis and evaluation of Alzheimer's Disease (Jack et al., 2024). These details are now explicitly reported in the updated

Table 1. The clinical consensus diagnosis was based on MACC's procedures, which consider performance on the Vascular Dementia Battery (VDB), structural MRI findings, and relevant biomarkers. The VDB evaluates six cognitive domains (attention, language, verbal and visual memory, visuoconstruction, and visuomotor speed) and classifies domain impairment as "borderline" (failure in one domain-specific test but <50% of tasks impaired) or "impaired" ($\geq 50\%$ of tasks failed). This battery has been validated in Singaporean stroke and dementia cohorts (Tham et al., 2002). NCI participants had Mini-Mental State Examination (MMSE) scores of ≥ 23 (secondary or tertiary education) or ≥ 21 (primary or no education), and no cognitive domain impairment on the VDB (Tham et al., 2002).

Etiological classification followed the NIA-AA recommendations, supported by plasma and neuroimaging biomarkers. Plasma p-tau217 levels, measured via SIMOA immunoassay, were used to stratify participants into low (≤ 0.388 pg/mL), intermediate (0.387–0.470 pg/mL), or high (≥ 0.471 pg/mL) risk of brain amyloid burden, consistent with thresholds validated in Singaporean cohorts (Chong et al., 2021, 2025). Participants underwent multimodal MRI scans, and significant cerebrovascular disease (CeVD) was defined as cortical infarcts and/or ≥ 2 lacunes and/or confluent white matter lesions in ≥ 2 regions (ARWMC ≥ 8 ; Hilal et al., 2015)

Accordingly, the cohort included both AD-only and mixed-pathology cases, reflecting the clinical heterogeneity typical of real-world memory clinic populations. A detailed investigation of vascular-specific navigation impairments is beyond the scope of the present manuscript and will form part of our future research with SPACE. Nonetheless, as suggested by the reviewer, we have conducted a series of exploratory robust ANOVA's to investigate differences in performance in the various SPACE tasks between patients with AD Non-Vascular, AD Vascular, Non-AD Non-Vascular, and Non-AD Vascular. Results revealed significant differences for training ($F_{3, 47.6} = 10.80, p = .001$), path integration ($F_{3, 26.5} = 4.57, p = .010$) and the perspective taking ($F_{3, 54} = 8.50, p < .001$) tasks. Critically, post hoc comparisons showed that Non-AD Non-Vascular patients performed significantly better than Non-AD Vascular patients for each of these tasks (all $ps \leq .038$).

We have edited the Methods section of the manuscript to reflect these changes:

Methods: Participants

The study included 300 participants ($M_{age} = 74$ years; 41% male). Patients were recruited from the memory clinics at the National University Hospital (NUH) and St. Luke's Hospital in Singapore, while control participants were recruited from these sites (90%) and the community (10%). Participants recruited from memory clinics were part of ongoing cohorts, i.e. HARMONISATION (Lim et al., 2025) and SINGER (Xu et al., 2022), at the Memory, Aging & Cognition Centre (MACC). All recruited participants underwent an initial telephone screening to assess their eligibility for the study. Participants were eligible if they were over 50 years old and could walk 10 meters without using a walking aid. Participants were excluded if they manifested severe visual impairment or hearing loss, a history of seizure, epilepsy, or acute cardiac events. Signed informed consent was obtained from all participants or their legal representative prior to undergoing the experimental procedure. Ethical approval for this study was provided by the NHG Domain Specific Review Board (DSRB) in Singapore (2021/01160).

Global Clinical Dementia Rating Diagnosis

The global Clinical Dementia Rating (CDR) score was determined relative to six cognitive and functional domains: memory, orientation, judgment and problem solving, community affairs, home and hobbies, and personal care. The CDR was calculated using the CDR-assignment algorithm (Morris, 1993). The CDR for each participant was either obtained from existing study records conducted within one year of the SPACE visit or reconstructed from comprehensive clinical notes and neuropsychological data when these records were unavailable. This method has been shown to achieve good to excellent agreement (intraclass correlation coefficient = 0.81–0.92)

with face-to-face assessments (Dauphinot et al., 2024). CDR scores were used to classify impairment severity: 0 = no dementia, 0.5 = questionable dementia, 1 = mild dementia, 2 = moderate dementia, and 3 = severe dementia (Table 1). Due to the limited number of participants with a CDR score of 3 ($n = 3$), the moderate and severe groups were merged for analysis.

Clinical Consensus Dementia Diagnosis

In addition to CDR classification, patients received a diagnosis by clinical consensus based on MACC's procedures, which consider performance on the Vascular Dementia Battery (VDB), structural MRI findings, and relevant biomarkers. The VDB evaluates six cognitive domains (attention, language, verbal and visual memory, visuoconstruction, and visuomotor speed) and classifies domain impairment as "borderline" (failure in one domain-specific test but $<50\%$ of tasks impaired) or "impaired" ($\geq 50\%$ of tasks failed). This battery has been validated in Singaporean stroke and dementia cohorts (Tham et al., 2002). Control participants had Mini-Mental State Examination (MMSE) scores of ≥ 23 (secondary or tertiary education) or ≥ 21 (primary or no education), and no cognitive domain impairment on the VDB. The diagnosis classified participants into groups reflecting cognitive severity (NCI = No Cognitive Impairment, SCD/SCI = Subjective Cognitive Decline/Impairment, MCI = Mild Cognitive Impairment, and Dementia), following the recommendation of the National Institute on Aging and the Alzheimer's Association (NIA-AA) for the diagnosis and evaluation of Alzheimer's Disease (Jack et al., 2024).

Etiological classification followed the recommendations of the NIA-AA and was supported by plasma and neuroimaging biomarkers. Plasma p-tau₂₁₇ levels were quantified via SIMOA immunoassay and used to stratify participants into low (≤ 0.388 pg/mL), intermediate ($0.387-0.470$ pg/mL), or high (≥ 0.471 pg/mL) risk of amyloid pathology according to thresholds validated in Singaporean cohorts (Chong et al., 2021, 2025). A subset of participants ($n = 209$) underwent multimodal MRI, with cerebrovascular disease (CeVD) defined as cortical infarcts and/or ≥ 2 lacunes and/or confluent white matter lesions in ≥ 2 regions ($ARWMC \geq 8$; Hilal et al., 2015). Accordingly, the cohort included individuals with AD Non-Vascular, AD Vascular, Non-AD Non-Vascular, and Non-AD Vascular. This composition captures the diversity of pathologies encountered in clinical settings. For participants without research scans, medical history (including CT scans when available) was used to determine CeVD status.

5. A battery of psych assessments were used and APOE4 collected were these considered in this study? If so, what were the outcomes? Did you explore TMT, maze task and others with CDR categorisation and SPACE performance? Processing speed and executive function are key components to findings regarding training time/increased processing speed

We agree that both neuropsychological tests and APOE4 are essential for contextualising the diagnostic sensitivity of SPACE. While the main focus of this study is the validation of SPACE relative to CDR-defined severity levels, we have now conducted additional analyses using the available neuropsychological and APOE genotype data.

Neuropsychological correlates and diagnostic performance

To characterise SPACE relative to established cognitive measures, we compared its diagnostic accuracy with that of standard neuropsychological tests (i.e., MoCA, QDRS, TMT A/B, Maze task, Animal Fluency, and Digit Cancellation Test, Dual task) across all CDR stage contrasts. This comparison is now presented in the main manuscript (Figure 3). We also include a new Table in the Supplementary Information (Supplementary Table 4) with additional details of the comparison (i.e., AUC, DeLong p value, sensitivity and specificity). Across all comparisons, SPACE showed robust discrimination between CDR levels and performed comparably to established neuropsychological tests.

For CDR 0 vs 0.5, no neuropsychological test differed significantly from SPACE, and all showed modest accuracy (AUC \approx 0.60–0.73). When comparing CDR 0 vs 1, the MoCA outperformed SPACE ($p = .021$), whereas the QDRS ($p = .010$), Maze ($p = .010$), and Dual task ($p = .007$) performed significantly worse. In differentiating CDR 0.5 vs 1, SPACE again achieved high discrimination (AUC = 0.91) and was significantly better than the QDRS ($p = .003$), TMTA ($p = .025$), Maze ($p = .008$), and Dual task ($p = .004$). No significant differences were observed for CDR 0 vs 2+, 0.5 vs 2+, or 1 vs 2+, where all tasks, including SPACE, performed strongly (AUCs > 0.85). Overall, SPACE showed the same or better discrimination than most conventional tests, particularly for transitions from very mild to mild dementia, with AUCs above 0.90 in all but the earliest (CDR 0 vs 0.5) and most advanced (CDR 1 vs 2+) stage comparisons.

Supplementary Table 4 | AUC for each predictor across CDR contrasts. AUC values reflect model performance at the Youden-optimal cutoff for discriminating between CDR levels. P-values are derived from DeLong's test comparing each neuropsychological test to the SPACE model, with Bonferroni correction applied across predictors within each CDR contrast.

Outcome	SPACE	MoCA	QDRS	TMTA	TMTB	Maze task	DCT	Animal	Dual task
	AUC, p (sensitivity, specificity)								
CDR 0 vs 0.5	0.61, Ref (0.47-0.79)	0.73, 0.169 (0.51-0.86)	0.69, 1.000 (0.66-0.63)	0.65, 1.000 (0.46-0.78)	0.65, 1.000 (0.38-0.88)	0.60, 1.000 (0.49-0.71)	0.62, 1.000 (0.52-0.74)	0.66, 1.000 (0.36-0.94)	0.60, 1.000 (0.54-0.66)
CDR 0 vs 1	0.94, Ref (1.00-0.85)	1.00, 0.021 (1.00-0.98)	0.80, 0.010 (0.70-0.77)	0.87, 0.284 (0.82-0.81)	0.94, 1.000 (0.85-0.90)	0.81, 0.010 (0.82-0.75)	0.87, 0.247 (0.77-0.81)	0.96, 1.000 (0.85-0.97)	0.79, 0.007 (0.62-0.84)
CDR 0 vs 2+	0.95, Ref (0.94-0.88)	1.00, 0.163 (1.00-1.00)	0.81, 0.168 (0.90-0.69)	0.96, 1.000 (0.94-0.89)	0.98, 1.000 (0.94-0.97)	0.90, 1.000 (0.71-0.94)	0.96, 1.000 (0.94-0.88)	1.00, 0.213 (1.00-0.98)	0.85, 0.989 (0.77-0.87)
CDR 0.5 vs 1	0.91, Ref (0.95-0.78)	0.93, 1.000 (1.00-0.74)	0.70, 0.003 (0.67-0.68)	0.76, 0.025 (0.88-0.57)	0.82, 0.405 (0.82-0.70)	0.73, 0.008 (0.85-0.67)	0.79, 0.098 (0.94-0.52)	0.86, 1.000 (0.85-0.74)	0.71, 0.004 (0.91-0.45)
CDR 0.5 vs 2+	0.91, Ref (0.88-0.85)	0.99, 0.266 (1.00-0.96)	0.81, 1.000 (0.90-0.65)	0.91, 1.000 (0.94-0.75)	0.91, 1.000 (0.81-0.88)	0.85, 1.000 (0.71-0.89)	0.92, 1.000 (0.82-0.88)	0.97, 0.892 (0.94-0.94)	0.82, 1.000 (0.69-0.85)
CDR 1 vs 2+	0.83, Ref (0.69-0.91)	0.92, 1.000 (0.82-0.88)	0.86, 1.000 (0.80-0.85)	0.85, 1.000 (0.77-0.85)	0.77, 1.000 (0.75-0.71)	0.83, 1.000 (0.71-0.94)	0.85, 1.000 (0.94-0.59)	0.86, 1.000 (0.71-0.88)	0.77, 1.000 (0.69-0.88)

Figure 3 | Forest plots showing the classification performance (AUC \pm 95% CI) of the SPACE tasks and neuropsychological tests across pairwise CDR group comparisons. Each panel represents a distinct comparison. Circles denote models including demographic variables (age, gender, education), while squares denote models excluding demographic variables (test performance only). Orange circles represent SPACE, and dark grey/blue markers represent traditional neuropsychological tests. Horizontal lines indicate 95% confidence intervals for the models, including demographics.

APOE genotype

APOE genotype data were only available for a subset of participants ($n = 96$) with unbalanced groups for genotypes ($\epsilon 2/\epsilon 2 = 3$; $\epsilon 2/\epsilon 3 = 9$; $\epsilon 3/\epsilon 3 = 57$; $\epsilon 3/\epsilon 4 = 25$; $\epsilon 4/\epsilon 4 = 2$). To facilitate analysis, participants were grouped as $\epsilon 4$ -carriers vs non-carriers. Results of a Mann-Whitney U test revealed no significant differences between groups for the training ($p = .460$), path integration ($p = 0.602$), pointing ($p = 0.886$), mapping ($p = 0.710$), and perspective taking ($p = 0.770$) tasks. Given that APOE genotype data were only available for a limited subset of participants, we excluded them from the main analysis and reported the results as an exploratory analysis in the Supplementary Information I (Supplementary Figure 2).

Supplementary Figure 2 | Performance across the tasks in SPACE for $\epsilon 4$ carriers vs non-carriers. Violin plots for the performance in the tasks in SPACE between APOE4 carriers and non-carriers, with embedded box plots and individual data points. No significant group differences were detected.

We have made the following changes in the manuscript:

Methods: Study Procedure

Upon arriving at the study site, participants received an overview of the study procedures. Once written informed consent was obtained, participants completed a series of questionnaires to collect sociodemographic data, self-reported navigation ability, and health information. Participants then completed a battery of neuropsychological tests and SPACE. The neuropsychological tests included the MoCA, QDRS, Trail Making Test A and B (TMT-A/B), Maze Task, Animal Fluency, Digit Cancellation Test (DCT), and a Dual Task measure of divided attention. After completing SPACE, blood ($n = 300$) and saliva samples ($n = 96$) were collected for biomarker analyses. Participants also completed a short gait assessment using single and dual tasks. The entire testing procedure lasted approximately 3 hours. This study is part of a wider data collection effort to validate SPACE in community and clinical samples.

Methods: Statistical Analysis

To compare the performance of SPACE relative to traditional neuropsychological tests, we conducted ROC analyses for each CDR stage contrast. Differences in AUC values between SPACE and other neuropsychological tests were examined using the DeLong test with Bonferroni correction. The same analysis was conducted to assess whether SPACE can discriminate between categories from the consensus diagnosis (NCI vs. Dementia, NCI vs. MCI, and MCI vs. Dementia).

[...]

Since the primary focus of this paper is to assess the performance of SPACE relative to the CDR scores, various other measures were excluded from the main analyses. A complete list of all measures collected and additional analyses is reported in the Supplementary Information.

Results

To compare the diagnostic accuracy of SPACE relative to standard neuropsychological measures, ROC analyses were performed, and differences in AUCs were assessed using Bonferroni-corrected DeLong test (see Figure 3 and Supplementary Table 4). For CDR 0 vs 0.5, no neuropsychological test differed significantly from SPACE, and all showed modest accuracy ($AUC \approx 0.60\text{--}0.73$). When comparing CDR 0 vs 1, MoCA outperformed SPACE ($p = 0.021$), whereas the QDRS ($p = 0.010$), Maze ($p = 0.010$), and Dual Task ($p = 0.007$) performed significantly worse. In differentiating CDR 0.5 vs 1, SPACE again achieved high discrimination accuracy ($AUC = 0.91$) and was significantly better than the QDRS ($p = 0.003$), TMT-A ($p = 0.025$), Maze ($p = 0.008$), and Dual Task ($p = 0.004$). No significant differences were observed for CDR 0 vs 2+, 0.5 vs 2+, or 1 vs 2+, where all tasks, including SPACE, performed strongly ($AUCs > 0.85$). Overall, SPACE showed the same or better discrimination than most conventional tests, particularly for transitions from very mild to mild dementia, with AUCs above 0.90 in all but the earliest (CDR 0 vs 0.5) and most advanced (CDR 1 vs 2+) stage comparisons.

Discussion

Comparing SPACE to standard neuropsychological tests allowed for direct benchmarking of its diagnostic validity. ROC analyses and DeLong tests confirmed that SPACE performed on par with, or better than, most conventional neuropsychological tests across clinical contrasts. Critically, for contrasts distinguishing CDR 0 vs 1 and 0.5 vs 1, SPACE showed comparable performance to the MoCA, TMT-B, and DCT, and outperformed the QDRS, TMT-A, Maze Test, and Dual Task. This pattern suggests that SPACE engages executive, attentional, working memory and processing speed components similar to those tapped by these tasks. The strong performance of the MoCA across all CDR stages reinforces its recognised sensitivity for detecting dementia. This strong performance was expected given the MoCA's broad domain coverage and the fact that part of the reconstructed CDR scores were derived from clinical notes and neuropsychological tests that include the MoCA. In this context, SPACE's ability to achieve AUCs close to those of the MoCA is particularly noteworthy.

- 6. The paper would benefit from more discussion about significant domains identified of SPACE sig group differences were found in training time ($F(3, 75\ 296) = 14.7, p < 0.001, \xi = 0.657$), path integration distance error ($F(3, 296) = 7.0, p = 0.004, \xi = 0.587$), and perspective taking error ($F(3, 296) = 11.0, p < 0.001, \xi = 0.51$). No significant group differences were observed for egocentric pointing error ($F(3, 296) = 2.61, p = 0.101$) or mapping accuracy - how do you interpret this? Further clarity on these domains and associated neuroanatomy/potential deficits – is this a strictly allocentric/path integration deficit? Does perspective taking tap into egocentric processing also?**

The reviewer asked us to clarify the lack of significant differences in egocentric pointing error and mapping accuracy. We believe that the lack of differences in these tasks can potentially be explained by their dependence on the path integration task. Both the pointing and mapping tasks rely on participants successfully encoding the position of the various landmarks when completing the various path integration trials. Here, participants who fail to accurately encode the locations of the landmarks are likely to perform worse on the path integration task and the following tasks (i.e., pointing and mapping) that depend on this knowledge. In contrast, performance on the perspective taking task does not depend on prior landmark encoding, as participants are provided with a map with the landmark locations during the task. To further investigate this relationship, we conducted an exploratory

analysis by performing a median split on path integration distance (median = 205 m) and computed Spearman's correlations separately for good and poor performers (Supplementary Figure 1).

Supplementary Figure 1 | Exploratory correlations between SPACE task performance in participants with good versus poor path integration ability. Spearman's rank correlations were computed separately for participants below (left panel) and above (right panel) the median path integration distance error (median = 205 m).

For participants who performed well in path integration, distance error correlated significantly with both pointing error ($\rho = 0.19$, $p = 0.038$) and mapping accuracy ($\rho = -0.29$, $p < 0.001$). In contrast, among those with poor path integration performance, no such correlations were found for pointing ($\rho = 0.04$, $p = 0.68$) and mapping ($\rho = -0.08$, $p = 0.399$). As expected, path integration error was correlated with perspective taking error for good ($\rho = 0.23$, $p = 0.009$) and bad performers ($\rho = 0.20$, $p = 0.026$). Together, these results support the interpretation that deficits in path integration can propagate to other tasks.

To further examine this relationship, we conducted separate robust regressions using path integration distance error and CDR level as predictors, and pointing error and mapping performance outcome variables, respectively. Here, we used robust regressions instead of robust ANCOVAs because the robust ANCOVA function in R package WRS2 only supports ANCOVAs with two groups, whereas our dataset included three CDR levels (0, 0.5, and 1). For pointing error, both path integration distance error ($\beta = 0.0266$, $p = 0.048$) and CDR 1 ($\beta = 10.73$, $p = 0.038$) were significant predictors. A similar pattern emerged for mapping performance, where both path integration distance error ($\beta = -0.00066$, $p = 0.008$) and CDR 1 ($\beta = 0.164$, $p = 0.038$) were significant. Specifically, higher path integration errors predicted poorer performance in pointing and mapping, while participants with CDR 1 showed reduced accuracy in both tasks compared to those with CDR 0.

We have conducted these additional analyses to address the reviewer's very pertinent comments, which are spot on! However, including pointing and mapping in the AUC models in the manuscript would be methodologically inappropriate, as these variables were not significant in the original classification analyses and share substantial variance with path integration error. Their post hoc inclusion would likely add unnecessary complexity without improving diagnostic accuracy and potentially increase the risk of collinearity and overfitting. Indeed, after adding these tasks and recalculating the AUCs, we found that they do not significantly alter the results (see Supplementary Table 7).

The new analysis and table are now included in the Supplementary Information G.

Supplementary Table 7 | Comparison of AUC values obtained with and without Pointing and Mapping (PM) measures across CDR group contrasts. The table reports the difference in AUC (Δ AUC), DeLong test p -values, and Bonferroni-corrected p -values for each comparison. Comparisons involving CDR = 2+ were omitted due to low variability in this subgroup.

CDR contrasts	AUC with PM	AUC No PM	Δ AUC	p DeLong	p Bonferroni
0 vs 0.5	0.603	0.613	-0.011	0.4568	1.0000
0 vs 1	0.964	0.945	0.019	0.1224	0.3672
0.5 vs 1	0.952	0.914	0.038	0.1330	0.3990

We have expanded the Discussion to further shed light on the relationship between the tasks and their ability to predict CDR group differences.

While the absence of significant group differences in the pointing and mapping tasks might initially appear inconsistent, these findings can be understood by considering their dependence on the path integration task. Both the pointing and mapping tasks rely on participants successfully encoding the position of the various landmarks when completing the various path integration trials. Here, participants who fail to accurately encode the locations of the landmarks are likely to perform worse on the path integration task and the following tasks (i.e. pointing and mapping) that depend on this knowledge. In contrast, performance on the perspective taking task does not depend on prior landmark encoding, as participants are provided with a map of the landmark locations. To clarify the relationship between these tasks, we conducted additional regression analyses examining whether clinical status and path integration performance jointly explained variation in pointing and mapping. These models showed that individuals who made larger errors during path integration also tended to perform worse on the pointing and mapping tasks, suggesting that difficulties in encoding landmark positions carried over into these later measures (Supplementary Information F).

In terms of the neuroanatomy associated with these potential deficits, we propose that the SPACE tasks lie along a continuum from egocentric to allocentric processing, rather than representing discrete points at either end. While path integration in SPACE is a primarily egocentric task that relies on the updating of position and orientation based on vection (illusory sensation of self-motion), it can also provide the self-motion framework from which allocentric representations, such as the relationships between landmarks, can emerge (Ekstrom et al., 2014). The pointing task requires participants to retrieve and transform this self-referenced information to estimate the direction of unseen landmarks, engaging both posterior parietal regions for egocentric updating (e.g., the waterfall is to my right) and retrosplenial regions for coordinate transformations (e.g., the waterfall is northeast of the rocket; Schinazi et al., 2013; Tu et al., 2015). The mapping task further demands integrating multiple viewpoints into a coherent representation of landmark locations, a process that relies on hippocampal and parahippocampal structures associated with allocentric spatial memory (Ruggiero et al., 2020; Wolbers & Buchel, 2005). However, the mapping task can technically also be solved procedurally by reconstructing the individual paths to each of the landmarks, potentially engaging the basal ganglia (Iaria et al., 2003; Weisberg et al., 2014). Finally, the perspective taking task involves mentally adopting a viewpoint other than one's own and judging where landmarks are from that imagined perspective (Kozhevnikov & Hegarty, 7 2001). The task draws on allocentric representations to reconstruct spatial relations and egocentric simulation to visualise the scene from the imagined perspective, reflecting a dynamic integration of both reference frames and engaging hippocampal and retrosplenial regions that enable flexible spatial transformations. Together, these tasks illustrate a somewhat hierarchical but not absolute shift from self-based updating to world-based representation that underpins spatial navigation (Lambrey et al., 2012; Schinazi et al., 2013). Because the tasks engage overlapping egocentric and allocentric processes, they cannot be strictly separated, and it is therefore not entirely possible to localise potential deficits to a single neural

system (Ekstrom et al., 2014). Future studies integrating neuroimaging with experimental designs optimised to disentangle egocentric and allocentric demands will be essential for identifying the neural substrates of these processes and refining the interpretability of deficits observed in SPACE.

We have expanded the Discussion to clarify the observed pattern of results across the SPACE tasks and to better situate them within the neurocognitive framework of spatial reference frames.

The tasks in SPACE were designed to engage a distributed network of brain regions along a continuum from egocentric to allocentric processing. Training engages visuospatial processing and motor control from an egocentric perspective supported by the posteromedial parietal (e.g., precuneus, posterior cingulate) and occipital regions (Cavanna & Trimble, 2006; Ilardi et al., 2022). Path integration is a primarily egocentric task that relies on the updating of position and orientation based on vection (illusory sensation of self-motion), which relies on the medial entorhinal cortex. This region interacts with the hippocampus, retrosplenial and parahippocampal cortices, medial prefrontal regions, and cerebellum, in order to integrate optic-flow, vestibular, and “proprioceptive” self-motion cues to maintain spatial orientation (Epstein et al., 2017; Schinazi & Thrash, 2018; Segen et al., 2022). However, path integration can also provide the self-motion framework from which allocentric representations, such as the relationships between landmarks, can emerge (Arnold et al., 2014; Chrastil et al., 2017; Ekstrom et al., 2014). The pointing task requires participants to retrieve and transform this self-referenced information to estimate the direction of unseen landmarks, engaging both posterior parietal regions for egocentric updating (e.g., the waterfall is to my right) and retrosplenial regions for coordinate transformations (e.g., the waterfall is northeast of the rocket; Schinazi et al., 2013; Tu et al., 2015). The mapping task further demands integrating multiple viewpoints into a coherent representation of landmark locations, a process that relies on hippocampal and parahippocampal structures associated with allocentric spatial memory (Ruggiero et al., 2020; Wolbers & Buchel, 2005). However, the mapping task can technically also be solved procedurally by reconstructing the individual paths to previously visited landmarks supported by the basal ganglia (Iaria et al., 2003; Weisberg et al., 2014). Finally, the perspective taking task involves mentally adopting a viewpoint other than one’s own and judging where landmarks are from that imagined perspective. This task draws on allocentric representations to reconstruct spatial relations and egocentric simulation to visualise the scene from the imagined perspective. This process reflects a dynamic integration of both reference frames, engaging hippocampal and retrosplenial regions that enable flexible spatial transformations (Lambrey et al., 2012; Schinazi et al., 2013). Together, these tasks illustrate a somewhat hierarchical but not absolute shift from self-based updating to world-based representation that underpins spatial navigation (Ekstrom et al., 2014).

- 7. Task appears very similar to domains tapped in spatial app Sea Hero Quest, consider how this literature/and others may inform/complement/make findings novel - G. Coughlan, A. Coutrot, M. Khondoker, A. Minihane, H. Spiers, & M. Hornberger, Toward personalized cognitive diagnostics of at-genetic-risk Alzheimer’s disease, Proc. Natl. Acad. Sci. U.S.A. 116 (19) 9285-9292, <https://doi.org/10.1073/pnas.1901600116> (2019).**

We thank the reviewer for this valuable comment. The study by Coughlan et al. (2019) is already cited in the Introduction within the sentence:

“Indeed, an accumulating body of evidence from real-world²⁰⁻²² and Virtual Reality (VR)²³⁻²⁸ studies has identified spatial navigation deficits as a promising marker of genetic risk for sporadic AD (APOE ε4-carriers)²⁹⁻³¹. ”

In this context, we specifically acknowledged Sea Hero Quest (SHQ) as a seminal contribution to the field, demonstrating how digital navigation tasks can serve as sensitive indicators of genetic risk for Alzheimer’s disease at the population level. However, we agree that the alignment of aims between SHQ and SPACE – both of which seek to capture individual differences in spatial navigation abilities as a proxy for cognitive impairment – merits

more explicit discussion. We have therefore expanded the Discussion to articulate how SPACE extends the contributions of SHQ and other virtual spatial navigation tests by validating similar concepts within a clinical framework and, through its shortened version (sSPACE), showing potential for large-scale, in-community deployment akin to SHQ.

We have expanded the Discussion to include contributions to the field of SHQ, other spatial navigation virtual assessments, and SPACE.

Discussion

Our findings build on prior work that has demonstrated that spatial navigation ability, assessed through digital tools, can be a sensitive marker of cognitive impairment (Berron et al., 2024; Chan et al., 2016; Coutrot et al., 2018; Liew et al., 2025; Meier et al., 2021; Rekers & Finke, 2024; Tu et al., 2015; van der Ham et al., 2020). Unlike traditional assessments, digital tools can enhance engagement and adherence (Hamari et al., 2014; Hampel et al., 2022; McLearnay, 2016; Polk et al., 2025), making them especially suitable for testing older adults in supervised and unsupervised settings. The pioneering citizen-study with Sea Hero Quest revealed the potential for scalable population-level screening, although its clinical relevance remains to be established (but see, Coughlan et al., 2019). Other virtual paradigms, such as the Virtual Supermarket Test (Tu et al., 2015) and the Four Mountains Test (Chan et al., 2016), have successfully discriminated between cognitively impaired and healthy individuals (Tu et al., 2015) and shown potential to predict conversion from MCI to dementia (Wood et al., 2016). However, these tools have been mainly confined to controlled laboratory or research contexts. SPACE aims to unify these two strands by translating the strengths of population-scale and laboratory-based approaches into a clinically validated tool benchmarked against standard neuropsychological assessments. Here, the shorter administration times and similar diagnostic accuracy in sSPACE make it ideal for future large-scale implementation in clinics and unsupervised settings.

Reviewer #2:

Thank you for letting me review this interesting article investigating a new cognitive task for detecting cognitive impairment in dementia. Overall, this is a well-conducted study in a large sample size, showing good discrimination of cognitive impairment and healthy. However, there are a few issues with the study that might warrant further revision.

1. **Overall, the clinical characterisation of the cohort is poor, considering that this is meant to detect cognitive impairment in dementia. There is no information provided on which clinical diagnoses people had. Also, the authors should provide the diagnostic criteria that were applied to the sample and whether the diagnoses were biomarker confirmed (it seems this data was collected but not presented).**

We thank the reviewer for this constructive comment, which echoes Comment 1 and Comment 4 from Reviewer 1. We agree that providing a more detailed description of the clinical characterisation, diagnostic criteria, and biomarker confirmation strengthens the manuscript and clarifies the clinical context of the study. We have substantially revised and extended Table 1, which now presents detailed information on the cohort's diagnostic composition, aetiology, vascular status, biomarker profiles, and performance on both neuropsychological and SPACE measures (see Comment 1, Reviewer 1). Moreover, we now clarify that all patients were diagnosed according to cognitive severity and aetiology during clinical consensus meetings held at the Memory, Aging & Cognition Centre, which involved the principal investigator, clinicians, and research staff. Diagnostic decisions were based on comprehensive clinical records, neuropsychological testing (Vascular Dementia Battery, VDB), neuroimaging, and blood biomarker profiles (see Comment 4, Reviewer 1).

2. **Along the same lines, over 80% of the sample has a CDR score of 0 or 0.5, which is considered to be healthy or having only very minor cognitive changes, not qualifying for a diagnosis of dementia. This large discrepancy between a large healthy group and people who have cognitive impairment (CDR >1) clearly questions in how far the cognitive impairment group was representative.**

We acknowledge that the sample is disproportionately biased towards CDR 0 (51%) and CDR 0.5 (32%). However, 17% of our sample still consists of individuals with a CDR of 1 or higher. In our sample, this proportion corresponds to approximately 24% MCI and 18% dementia (based on clinical consensus), which aligns with the distribution of cognitive impairment within the global geriatric population. Indeed, systematic reviews have shown that the global prevalence of MCI can range from 15% (Bai et al., 2022) to 23% (Salari et al., 2025). Similar distributions have also been reported in recent large-scale studies that develop and validate digital cognitive tools (Berron et al., 2024; Liew et al., 2025). For example, Liew and colleagues' (2025) sample included 11% MCI and 2% dementia (CDR 0 = 85%; CDR 0.5 = 15%; CDR \geq 1 = 2%), while Berron and colleagues' (2024) sample included 18% MCI and no dementia cases (no CDR data is available for this dataset).

We have revised the Discussion to acknowledge the limitations associated with the class imbalance while contextualising our sample within the broader literature. Here, we highlight that its composition is consistent with the demographic reality of cognitive impairment in ageing populations and with comparable large-scale digital assessment studies.

Discussion

[...] Third, although the proportion of MCI cases is consistent with prevalence estimates in an ageing population (Bai et al., 2022; Salari et al., 2025) and large-scale studies on digital assessment (Berron et al., 2024; Liew et al., 2025), the

sample was nonetheless skewed toward individuals with CDR 0–0.5, with relatively few participants having a CDR > 1. This imbalance may limit the representativeness of the cognitively impaired group.

3. **It is not clear which cognitive profile/deficits the cohort has, since there is no other cognitive data provided. There should be at least a cognitive screening tests, e.g. MoCA, or even better the cohort should have been characterised by neuropsychological testing, to determine which cognitive deficits they had. Such other cognitive testing would have also allowed how the SPACE battery compares to existing gold standard cognitive testing, which is in the current version not clear. The cognitive impairment is solely based on the CDR, which is a clinical assessment and not a cognitive assessment per se.**

We thank the reviewer for this comment. As detailed in our responses to Reviewer #1, we have already included a full characterisation of the cohort (Comment 1, Reviewer 1) and additional analyses comparing SPACE with standard neuropsychological assessments (Comment 5, Reviewer 1). Specifically, we benchmarked SPACE against the MoCA, QDRS, TMT A/B, Maze Task, Animal Fluency, Digit Cancellation Test, and Dual Task, assessing their respective AUCs, sensitivities, specificities, and Bonferroni-corrected p-values derived from DeLong’s tests across all CDR contrasts. Results reveal that SPACE achieved comparable or superior diagnostic performance relative to most conventional measures (Figure 3 and Supplementary Table 4). SPACE reached an AUC of 0.94 for CDR 0 vs 1 and an AUC of 0.91 for CDR 0.5 vs 1, performing on par with the MoCA and TMT-B while significantly outperforming several other measures (e.g., QDRS, TMT-A, Maze Task, Dual Task). No significant differences were observed between SPACE and any comparator in the majority of other contrasts ($p \geq 0.163$). Here, it is worth highlighting that the MoCA (AUC = 1.00) outperformed SPACE (AUC = 0.94) only for the CDR 0 vs CDR 1 contrast.

To further demonstrate that SPACE is psychometrically aligned with the best (based on our results) neuropsychological test, we conducted a hierarchical regression analysis (Supplementary Table 6) with MoCA as the outcome variable and the tasks in SPACE as predictors while controlling for key demographic variables (age, gender, and education). The model accounted for a substantial amount of the variance in MoCA ($F_{(12,241)} = 20.50$, $p < .001$, $R^2 = 0.505$) after accounting for demographic variables ($\Delta R^2 = 0.314$, $p < .001$).

Table 6 | Model fit indices and comparison between hierarchical regression models to predict MoCA.

Model	R	R ²			
1	0.437	0.191			
2	0.711	0.505			
Model comparison	ΔR^2	F	df1	df2	p
Model 1 – 2	0.314	25.5	6	241	<.001
Predictor	Estimate	SE	t	p	
Intercept ^a	35.80144	3.38906	10.5638	<.001	
Age	0.03654	0.03517	1.0390	0.300	
Gender: Female - Male	0.92017	0.41705	2.2064	0.028	
Education: Primary – No formal education	-3.23151	1.22610	-2.6356	0.009	
Education: Secondary – No formal education	-0.81176	1.15777	-0.7011	0.484	
Education: High school – No formal education	0.08665	1.16639	0.0743	0.941	
Education: University – No formal education	0.96165	1.19181	0.8069	0.421	
Training Time	-0.03321	0.00450	-7.3821	<.001	
PI Distance	-0.00789	0.00235	-3.3564	<.001	
Pointing Error	-0.04404	0.01120	-3.9315	<.001	

Predictor	Estimate	SE	t	p
Map R ²	-1.69874	0.67898	-2.5019	0.013
Memory Correct	0.04065	0.00935	4.3477	<.001
Perspective Error	-0.04344	0.01068	-4.0662	<.001

In response to the comment that cognitive impairment in our study was defined solely by the CDR, which is a clinical rather than a cognitive assessment, we would like to clarify that in both clinical research and practice, diagnostic classification and staging of dementia are primarily based on comprehensive clinical evaluation using the CDR. The CDR remains the gold standard for clinical diagnosis in Singapore, where the experiment was conducted, because it integrates cognitive performance, informant reports, and functional status into a multidimensional rating that reflects real-world clinical judgment.

Nonetheless, we also compared SPACE relative to consensus diagnosis (NCI vs. Dementia, NCI vs. MCI, and MCI vs. Dementia). For each comparison, we calculated AUC values along with sensitivity and specificity at the optimal cut-off. The p-values reported in the table indicate the statistical tests comparing the AUC of each assessment with that of SPACE, which served as the reference model. Across the three diagnostic contrasts, SPACE demonstrated strong alignment with clinical consensus (Supplementary Table 5). In distinguishing NCI from Dementia, SPACE showed excellent discrimination (AUC = 0.94). In this comparison, MoCA performed significantly better than SPACE (AUC = 1.00, $p = 0.006$), whereas QDRS performed significantly worse (AUC = 0.82, $p = 0.037$). All other tests showed no significant difference from SPACE. For NCI versus MCI, MoCA again outperformed SPACE (AUC 0.86 vs AUC 0.72, $p = .002$), while the remaining assessments were statistically indistinguishable from it. Finally, for MCI versus Dementia, none of the cognitive measures differed significantly from SPACE (see Figure 3 and Supplementary Table 5).

Supplementary Table 5 | Diagnostic accuracy of cognitive and digital assessments against clinical consensus diagnosis.

Outcome	AUC, p (Sensitivity, Specificity)							
	SPACE	MoCA	QDRS	TMTA	TMTB	Maze	DCT	Animal
NCI vs Dementia	0.94, Ref (1.00, 0.85)	1.00, 0.006 (1.00, 1.00)	0.82, 0.037 (0.71, 0.80)	0.92, 1.000 (0.94, 0.80)	0.96, 1.000 (0.94, 0.88)	0.83, 0.053 (0.82, 0.77)	0.87, 0.593 (0.82, 0.78)	0.99, 0.063 (1.00, 0.92)
NCI vs MCI	0.72, Ref (0.60, 0.76)	0.86, 0.002 (0.77, 0.84)	0.74, 1.000 (0.63, 0.76)	0.77, 0.895 (0.67, 0.77)	0.81, 0.103 (0.62, 0.88)	0.74, 1.000 (0.75, 0.65)	0.78, 1.000 (0.68, 0.77)	0.80, 0.223 (0.65, 0.81)
MCI vs Dementia	0.87, Ref (0.94, 0.73)	0.91, 1.000 (1.00, 0.72)	0.71, 0.214 (0.71, 0.67)	0.73, 0.411 (0.71, 0.78)	0.76, 1.000 (0.82, 0.65)	0.72, 0.318 (0.65, 0.75)	0.73, 0.450 (0.76, 0.70)	0.90, 1.000 (0.88, 0.75)

We have edited the text in the manuscript which now read as follow:

Methods: Statistical Analysis

To compare the performance of SPACE relative to traditional neuropsychological tests, we conducted ROC analyses for each CDR stage contrast. Differences in AUC values between SPACE and other neuropsychological tests were examined using the DeLong test with Bonferroni correction. The same analysis was conducted to assess whether SPACE can discriminate between categories from the consensus diagnosis (NCI vs. Dementia, NCI vs. MCI, and MCI vs. Dementia).

Results

While the primary goal of this paper was to assess whether SPACE can discriminate between various CDR scores, we also compared it relative to consensus diagnosis. For each comparison, we calculated AUC values along with sensitivity and specificity at the optimal cut-off. The *p*-values reported in the table indicate the statistical tests comparing the AUC of each assessment with that of SPACE, which served as the reference model. For these analyses, we merged the No Cognitive Impairment (NCI) and Subjective Cognitive Decline/Impairment (SCD/SCI) groups into a single NCI category, as SCD/SCI reflects concerns about cognitive changes but does not constitute a formal clinical diagnosis of cognitive impairment. In distinguishing between NCI from Dementia, SPACE demonstrated excellent discrimination accuracy ($AUC = 0.94$). In this comparison, MoCA performed significantly better than SPACE ($AUC = 1.00$, $p = 0.006$), whereas QDRS performed significantly worse ($AUC = 0.82$, $p = 0.037$). All other tests showed no significant difference from SPACE. For NCI versus MCI, MoCA again outperformed SPACE ($AUC\ 0.86$ vs $AUC\ 0.72$, $p = .002$), while the remaining assessments were statistically indistinguishable from it. Finally, for MCI versus Dementia, none of the cognitive measures differed significantly from SPACE (see Supplementary Tables 5 and 6).

- 4. The authors show that the path integration condition is one of the most sensitive to detect cognitive impairment but then remove this condition from the short version of the test. This seems an odd decision, solely based on time constraints for testing. Would it not make sense to keep the most sensitive measures, regardless of time taken for testing? Please provide a more detailed rationale.**

We thank the reviewer for this insightful comment and the opportunity to clarify our rationale. We fully agree that path integration is a sensitive task for detecting cognitive impairment, and this is supported by a growing body of research (Bierbrauer et al., 2020; Castegnaro et al., 2023; Howett et al., 2019; Mokrisova et al., 2016; Newton et al., 2024). Indeed, this was one of the key findings of our study, underscoring the value of including this task in the full version of SPACE. The development of the short version SPACE (sSPACE), however, was guided by a different yet complementary objective: to create a brief screening tool that could be administered with ease at home or within the strict time constraints typical of memory clinics and community settings. We nevertheless recognise the reviewer's point that excluding path integration purely on the basis of time warrants further justification, particularly if its inclusion could meaningfully improve diagnostic performance.

To evaluate this, we compared AUCs for each CDR contrast in sSPACE with and without the path integration task. We excluded contrasts involving CDR 2 and above because participants in these groups frequently aborted the path integration task due to its length and complexity, opting instead to proceed directly to the perspective taking task. The addition of the path integration task did not significantly improve discrimination accuracy for CDR 0 vs CDR 0.5 ($p = 0.777$, $p_{\text{Bonferroni}} = 1.000$) and CDR 0 vs CDR 1 ($p = 0.107$, $p_{\text{Bonferroni}} = 0.322$). We found that including the path integration task significantly improved the AUC for CDR 0.5 and CDR 1 ($p = 0.019$), increasing it from 0.83 to 0.91. However, this comparison did not survive Bonferroni correction ($p_{\text{Bonferroni}} = 0.056$).

We have revised the Results section for sSPACE to include this additional analysis and discuss its implications in the Discussion.

Results

Given the known role of path integration in discriminating impaired from not-impaired individuals, we compared AUCs for each CDR contrast in sSPACE with and without the path integration task. Here again, we excluded contrasts involving CDR 2+ because participants in these groups frequently aborted the path integration task due to its length and complexity, opting instead to proceed directly to the perspective taking task. The addition of the path integration task did not significantly improve discrimination accuracy for CDR 0 vs CDR 0.5 ($p = 0.777$,

$p_{\text{bonferroni}} = 1.000$) and CDR 0 vs CDR 1 ($p = 0.107$, $p_{\text{bonferroni}} = 0.322$). We found that including the path integration task significantly improved the AUC for CDR 0.5 and CDR 1 ($p = 0.019$), increasing it from 0.83 to 0.91. However, this comparison did not survive Bonferroni correction ($p_{\text{Bonferroni}} = 0.056$).

Discussion

However, because path integration showed greater sensitivity for discriminating between CDR 0.5 and CDR 1, administering the full SPACE battery remains advisable whenever feasible.

- 5. When comparing AUC or model performance for mild and moderate, or normal and very mild, there is some inconsistency in the wording that sometimes suggests it is significant, while other times it is not. Please check this carefully throughout the manuscript.**

We thank the reviewer for this observation. We have carefully reviewed the manuscript for inconsistencies in wording and reporting of significance. All terminology has been revised to align with standard CDR classifications (0 = no dementia, 0.5 = questionable dementia, 1 = mild dementia, 2 = moderate dementia, 3 = severe dementia), and any inconsistencies in significance reporting have been corrected throughout the text.

- 6. The paragraph explaining SPACE classification performance for normal vs very-mild impairment needs to be incorporated earlier for better context moving forward. Also, the explanation given as to why the results might not be significant is insufficient. It would be better if the authors could find an explanation within the SPACE tasks themselves, instead of a very vague explanation.**

The paragraph describing SPACE's classification performance for normal vs. very-mild impairment has been moved earlier in the Discussion section to provide better context. We have also expanded the discussion with a more detailed explanation of why there were no differences in CDR levels for some tasks in SPACE (see Reviewer 1 Comment 6).

- 7. Finally, it is odd to state that "SPACE consistently improved model performance compared to sociodemographic variables alone." Usually, no classification would be only conducted on sociodemographic variables, since diagnostic, cognitive and functional variables usually determine this.**

The purpose of comparing SPACE to the demographic model (i.e., age, gender and education) is to show that SPACE adds predictive power beyond these variables. Indeed, this approach has been previously adopted in the literature (Berron et al., 2024). To complement this, we have now conducted parallel ROC analyses on standard neuropsychological tests that were part of the assessment, including the MoCA, QDRS, Trail Making Tests A/B, the Maze Task, Digit Cancellation Task, Animal Fluency, and Dual task to provide a direct comparison between SPACE and established measures of cognitive and functional performance (see Reviewer 1, Comment 5). These results have been added to the revised manuscript in Figure 3 and Supplementary Table 4 to further contextualise the diagnostic value of SPACE.

We have revised the text in the Methods to clarify that the purpose of this comparison was not to propose sociodemographic variables as an alternative diagnostic model, but to demonstrate that SPACE provides additional discriminative value above known risk factors.

Methods

The purpose of this comparison was to establish a baseline model using known demographic risk factors for dementia, against which the added discriminative value of SPACE performance could be evaluated.

Reviewer #4:

In this large clinimetric study, the authors evaluated the discriminatory power of the Spatial Performance Assessment for Cognitive Evaluation (SPACE), a novel and promising tablet-based tool for assessing spatial navigation. The tool appears impressive in distinguishing individuals with varying degrees of cognitive impairment. SPACE was tested in both a memory clinic and a community cohort, including 300 participants. I have carefully reviewed the manuscript. Below are my impressions and my suggestions.

Major concerns

From a clinimetric standpoint, the tablet-based paradigm devised by the authors shows impressive performance in terms of classification accuracy. Nevertheless, after careful examination of the manuscript, several concerns arise:

- 1. First, it is puzzling that, despite the availability of extensive neuropsychological and neurobiological data, patient classification remains rudimentary, with no reference to clinical phenotypes or etiological diagnoses. Patient classification indeed relies only on the CDR score.**

We thank the reviewer for this insightful comment, which also echoes points raised by Reviewers 1 and 2 regarding the need for a clearer description of the clinical composition of our sample. We agree that the original version of the manuscript did not sufficiently reflect the richness of the available neuropsychological and neurobiological data. To address this, we have substantially revised and extended Table 1, which now provides a comprehensive overview of participants' diagnostic classifications, aetiological subtypes, vascular comorbidities, and biomarker profiles (including APOE genotype and plasma-based markers, where available). We have also included summary data for their performance on standard neuropsychological tests and on SPACE tasks to facilitate comparison across domains. We believe these additions provide a complete and more transparent characterisation of the cohort and substantially strengthen the clinical interpretability of our findings. For more details, see our response to Comment 1 from Reviewer 1 and Comment 1 from Reviewer 2.

- 2. Second, why was the “SPACE” model not compared to a control/reference model incorporating standard cognitive measures, such as the MoCA?**

We thank the reviewer for this constructive comment, which also echoes similar feedback received from other reviewers. We fully agree that including standard cognitive measures provides an important benchmark for evaluating the discriminative validity of SPACE relative to established neuropsychological assessments. We also conducted a complementary analysis examining whether SPACE performance predicts MoCA scores while controlling for key demographic covariates (age, gender, and education). We refer the reviewer to Reviewer 1 Comment 5 and Reviewer 2 Comment 3. To anticipate, SPACE demonstrated comparable diagnostic accuracy to conventional tests such as the MoCA and TMT-B, achieving AUCs of up to 0.94 for CDR 0 vs 1 and 0.91 for CDR 0.5 vs 1, while significantly outperforming several other measures (e.g., QDRS, TMT-A, Maze Task, Dual Task). Furthermore, SPACE explained a substantial proportion of variance in MoCA scores ($F_{(12,241)} = 20.50, p < .001, R^2 = 0.505$), indicating strong convergent validity. These results are now summarised in Figure 3 and in Supplementary Information D and E.

- 3. Third, critical variables – such as tablet experience, depression, anxiety, and sleep – are neither reported nor screened for their eligibility as covariates within the authors' proposed models.**

We agree with the reviewer that it is important to report the effect of covariates on SPACE performance. To address this, we conducted additional separate robust regressions that included potential confounding variables (i.e., age, gender, education, depression, anxiety, chronic stress, sleep hours, prior tablet experience, and self-assessed navigation ability - SBSOD) as covariates and the SPACE tasks (i.e., training time, path integration distance error, pointing error, map r^2 , and perspective taking error) as outcome variables (Supplementary Table 10).

Results revealed that tablet experience and sleep hours were the only significant predictors for performance in some SPACE tasks. Specifically, tablet experience predicted training time ($p = 0.029$), while sleep hours were positively associated with path integration distance error ($p = 0.016$) and mapping ($p = 0.039$). Critically, CDR was not a significant predictor of pointing and mapping performance after adjusting for covariates. As expected, CDR level remained a robust predictor of performance for the training, path integration, and perspective taking tasks, even after adjusting for the covariates. We also conducted post-hoc Tukey-adjusted pairwise comparisons to identify the significant contrast (Supplementary Table 11). In the training task, participants with CDR 1 required significantly longer training times than CDR 0 ($p = .014$). In the path integration task, both the CDR 1 vs CDR 0 ($p = .007$) and CDR 1 vs CDR 0.5 ($p = .025$) contrasts were significant, indicating increased distance error with higher clinical severity. Similarly, perspective taking errors were significantly greater in the CDR 1 group relative to both CDR 0 ($p = .017$) and CDR 0.5 ($p = .004$).

Supplementary Table 10 / Robust linear regression models including depression, anxiety, chronic stress, sleep hours, prior tablet experience, and self-assessed sense of direction (SBSOD) as covariates across SPACE tasks.

Predictor	Training			Path integration			Pointing			Mapping			Perspective taking		
	Est	CI	p	Est	CI	p	Est	CI	p	Est	CI	p	Est	CI	p
(Intercept)	226.90	134.41 – 319.39	<0.001	61.23	-107.60 – 230.07	0.476	84.60	50.50 – 118.70	<0.001	0.58	-0.03 – 1.19	0.061	19.04	-25.34 – 63.42	0.399
CDRglobal [0.5]	10.40	-1.83 – 22.62	0.095	3.09	-15.75 – 21.93	0.747	0.61	-4.66 – 5.88	0.820	0.01	-0.08 – 0.10	0.819	-1.45	-6.34 – 3.44	0.561
CDRglobal [1]	33.19	9.65 – 56.73	0.006	62.69	21.35 – 104.04	0.003	9.40	-0.51 – 19.30	0.063	0.12	-0.02 – 0.27	0.088	10.19	2.86 – 17.53	0.007
Age	0.79	-0.19 – 1.78	0.115	2.59	1.28 – 3.89	<0.001	-0.01	-0.40 – 0.39	0.974	-0.00	-0.01 – 0.00	0.419	0.52	0.06 – 0.98	0.027
Gender [1]	7.98	-3.42 – 19.39	0.169	22.99	5.53 – 40.46	0.010	-0.36	-5.32 – 4.59	0.886	0.08	-0.00 – 0.17	0.059	0.72	-4.21 – 5.65	0.775
education4levels [1]	-7.96	-45.19 – 29.28	0.674	-80.90	-209.86 – 48.05	0.218	9.86	-5.24 – 24.96	0.200	-0.14	-0.39 – 0.12	0.292	-10.27	-28.49 – 7.94	0.268
education4levels [2]	-28.27	-60.11 – 3.57	0.082	-92.58	-220.03 – 34.86	0.154	5.87	-8.23 – 19.98	0.413	-0.04	-0.28 – 0.21	0.759	-9.94	-27.06 – 7.18	0.254
education4levels [3]	-30.53	-62.93 – 1.87	0.065	-82.33	-211.03 – 46.37	0.209	4.77	-9.54 – 19.08	0.512	0.01	-0.23 – 0.26	0.925	-13.42	-30.69 – 3.85	0.127
education4levels [4]	-19.57	-52.92 – 13.78	0.249	-94.64	-223.15 – 33.86	0.148	2.56	-11.80 – 16.93	0.726	0.01	-0.24 – 0.26	0.957	-18.85	-35.99 – -1.71	0.031
Anxiety	-1.88	-6.51 – 2.75	0.425	-2.10	-9.01 – 4.80	0.549	0.60	-1.63 – 2.84	0.596	0.02	-0.01 – 0.05	0.208	-0.92	-2.84 – 1.00	0.347
ChrStress	1.14	-3.20 – 5.49	0.605	5.00	-1.31 – 11.31	0.120	0.10	-1.98 – 2.18	0.925	-0.03	-0.06 – 0.00	0.070	1.35	-0.68 – 3.38	0.191
Depression	-0.64	-5.83 – 4.54	0.807	-0.53	-9.40 – 8.34	0.907	-0.97	-3.14 – 1.20	0.378	0.01	-0.02 – 0.05	0.389	0.51	-1.72 – 2.73	0.653
SleepHrs	1.33	-2.44 – 5.10	0.488	4.79	0.91 – 8.67	0.016	-0.26	-1.66 – 1.13	0.710	0.02	0.00 – 0.04	0.039	-1.06	-3.00 – 0.88	0.281
ExpTablet [1]	-9.34	-28.79 – 10.10	0.345	4.29	-26.59 – 35.18	0.784	-2.33	-9.96 – 5.30	0.548	-0.00	-0.13 – 0.12	0.959	-2.96	-10.61 – 4.70	0.448
ExpTablet [2]	-20.91	-39.68 – -2.14	0.029	-7.09	-37.47 – 23.29	0.646	-6.25	-14.02 – 1.52	0.114	0.07	-0.06 – 0.20	0.274	-6.93	-14.80 – 0.94	0.084
SBSOD	2.50	-2.80 – 7.81	0.354	-3.13	-10.41 – 4.14	0.397	-1.30	-3.74 – 1.14	0.294	-0.02	-0.06 – 0.02	0.383	2.48	-0.04 – 4.99	0.054
Observations	283			255			254			253			280		
R2 / R2 adjusted	0.225 / 0.182			0.307 / 0.264			0.073 / 0.014			0.086 / 0.028			0.211 / 0.167		

Table 11 | Robust pairwise post hoc contrasts (Tukey-adjusted) for the effect of clinical diagnosis (CDR) across SPACE task outcomes. Models were fitted with robust linear regression (*lmrob*) controlling for age, gender, education, depression, anxiety, chronic stress, sleep hours, tablet experience, and SBSOD. Significance codes: *** p < .001, ** p < .01, * p < .05, ns = not significant.

Outcome	Contrast (CDR)	Estimate	SE	z	p
Training Time	0.5 – 0	10.395	6.211	1.674	0.204 ns
	1 – 0	33.192	11.957	2.776	0.014 *
	1 – 0.5	22.797	12.688	1.797	0.161 ns
Path Integration Distance	0.5 – 0	3.090	9.564	0.323	0.940 ns
	1 – 0	62.695	20.989	2.987	0.007 **
	1 – 0.5	59.605	23.266	2.562	0.025 *
Perspective Error	0.5 – 0	-1.448	2.489	-0.582	0.827 ns
	1 – 0	10.189	3.736	2.727	0.017 *
	1 – 0.5	11.637	3.679	3.163	0.004 **

4. Similarly, data on fluid biomarkers and APOE status were not included in the analysis. Why?

We thank the reviewer for this important comment. At the time of initial manuscript submission, plasma samples were collected but had not yet been analysed. Therefore, we had intentionally reserved the analysis of fluid biomarkers for a future publication, which will be more specifically focused on biomarker–behaviour relationships. Since submission, plasma analyses have been completed, and these data have been incorporated into the updated diagnosis now presented in the revision.

Supplementary Table 12 | Binomial logistic regression coefficients predicting p-tau biomarker risk (low vs. high) from demographic and cognitive variables (Model 2) with model fit statistics.

Predictor	Estimate	SE	Z	p
Intercept	-9.41509	2.72868	-3.4504	<.001
Age	0.04216	0.02926	1.4409	0.150
Gender:				
1 – 0	-0.29944	0.34433	-0.8696	0.384
education4levels:				
1 – 0	1.40700	0.95123	1.4791	0.139
2 – 0	0.61074	0.93220	0.6552	0.512
3 – 0	1.01179	0.92674	1.0918	0.275
4 – 0	0.68573	0.95651	0.7169	0.473
Training time	0.01539	0.00364	4.2255	<.001
Path integration distance	3.81e-4	0.00186	0.2046	0.838
Pointing error	0.00232	0.00911	0.2549	0.799
Map R ²	-0.41268	0.55907	-0.7382	0.460
Perspective taking error	-6.40e-4	0.00877	-0.0730	0.942
Model	Deviance	AIC	R²_{McF}	
1	258	272	0.0560	
2	236	260	0.1347	
Model Comparison (1 vs. 2)	$\chi^2(5) = 21.5, p < .001$			

In response to the reviewer’s request, we conducted exploratory analyses using the available biomarker data to examine potential associations with task performance. Specifically, logistic regression models were run with p-tau biomarker risk (low vs. high) as the dependent variable and demographic variables (i.e., age, gender and education) and SPACE tasks as predictors. Training time emerged as a significant predictor ($\beta = -0.015, p < 0.001$), indicating an association between longer training times and higher p-tau risk. No significant effects were found for age, gender, education, or other task measures. With respect to APOE status, as noted in our response to Reviewer 1 Comment 5, no significant differences were observed between $\epsilon 4$ carriers and non-carriers across any of the SPACE tasks. We report the APOE results in the Supplementary Information but reserve a detailed analysis of blood-based biomarkers for a future, more focused publication, with the understanding of the editor and reviewers.

Minor concerns

- 1. Introduction:** In lines 28–32, the authors claim that multidomain cognitive impairment may become detectable, especially when the neurodegenerative disease reaches an advanced stage. This statement is not

necessarily true and should be more carefully contextualized. Clinically, the primary issue lies in the limited sensitivity of standard neuropsychological assessments, which often fail to capture early or subtle cognitive deficits. Furthermore, references 11 and 12 concern spatial navigation and do not support the authors' statement. I suggest consulting the following references for a more theoretical discussion of this crucial point.

Suggested references: Ilardi et al. J Alzheimers Dis. doi:10.3233/JAD-240339.

We thank the reviewer for highlighting this point. We have revised the paragraph in the Introduction to clarify that the limitation lies in the sensitivity of standard neuropsychological assessments to detect early or subtle cognitive deficits. Additional references (Hampel et al., 2022; Ilardi et al., 2024; Polk et al., 2025; Rentz et al., 2013; Watermeyer & Calia, 2019) have been added to support this clarification, while reference 11 (Coughlan et al., 2018) was retained as it further demonstrates that spatial deficits may serve as sensitive early marker of impairment.

The new paragraph in the Introduction now reads:

Episodic memory impairments have long been the hallmark of MCI and AD, forming the cornerstone of diagnostic criteria. Current neuropsychological assessments typically focus on the detection of memory deficits alongside impairments in other cognitive domains (e.g., attention, executive function). However, the limited sensitivity of some standard screening tools frequently hinders the detection of early or subtle cognitive changes, particularly in the preclinical stages of neurodegenerative disease (Coughlan et al., 2018; Hampel et al., 2022; Ilardi et al., 2024; Polk et al., 2025; Rentz et al., 2013; Watermeyer & Calia, 2019). As a result, many diagnoses are still made only after significant neurodegeneration has already occurred, when interventions are less effective. While blood biomarkers simplify early detection compared to Positron Emission Tomography (PET) and Cerebrospinal Fluid (CSF) methods, their reliance on clinic-based sampling and interpretation does not fully overcome the constraints of late-stage, in-clinic diagnosis.

2. **Methods: It is somewhat surprising that the study protocol includes a gait assessment but does not incorporate a comprehensive evaluation of upper extremity motor function.**

We included the description of the gait assessment in the manuscript to provide a transparent overview of the broader study protocol. However, the focus of the present work is on the cognitive assessment component rather than motor performance. The gait data were collected as part of a multimodal dataset to enable future analyses on the relationship between motor and cognitive function, which will be presented in a separate publication. While the gait protocol involved wearable sensors placed on the trunk, feet, and hands, the latter were primarily used to ensure synchronisation and balance of the overall measurement system rather than to derive detailed upper-limb motor parameters. Therefore, a comprehensive evaluation of upper-extremity motor function is not possible or within the scope of this manuscript.

3. **Results: group differences in spatial navigation. Why were group differences analyzed using a one-way ANOVA with bootstrapping? Is there a specific rationale behind this methodological choice?**

We used robust one-way ANOVAs with bootstrapping to ensure reliable inference under potential violations of classical parametric assumptions. Robust estimators (e.g., trimmed means) mitigate the influence of outliers and non-normality on central tendency measures. However, they still rely on theoretical sampling distributions that can be inaccurate when sample sizes are small, group variances are unequal, or data are not normally distributed. Bootstrapping complements the robust ANOVA by empirically approximating the sampling distribution of the test

statistic through repeated resampling of the observed data. This approach provides more accurate p-values and confidence intervals.

a) How did the authors compute the effect size (ξ)? Please, describe how this measure was quantified, as well as any conventional rules of thumb for interpreting its magnitude.

We appreciate the reviewer's request for clarification. The effect size reported (ξ , ξ_i) was computed automatically by the Robust ANOVA module in Jamovi, which implements procedures from the WRS2 R package (Mair & Wilcox, 2020). Here, ξ represents a robust analogue of eta-squared (η^2), quantifying the proportion of explained variance when the data are heteroscedastic or non-Gaussian. Unlike η^2 , which relies on mean squares from the classical ANOVA, ξ was calculated as the ratio of the variance of robust group location estimates (in our case, M-estimator) to a robust measure of total variance.

Following the interpretation guidelines proposed by Wilcox and Tian (2011), typical benchmarks for ξ are:

- Small effect: $\xi \approx 0.10$
- Medium effect: $\xi \approx 0.30$
- Large effect: $\xi \geq 0.50$

b) Table 2 is informative, but it is likely more appropriate to be placed in the supplementary materials. I recommend that the authors concisely summarize these results in the main text and present the between-group comparisons in a graphical format.

We thank the reviewer for this helpful suggestion. We agree that Table 2 is more appropriate for the Supplementary Information. Accordingly, we have moved it there (Supplementary Table 2), concisely summarised the results in the main text, and present a new figure with the between-group comparisons in the main manuscript.

Figure 1 | Between-group comparisons across CDR levels for (a) training time, (b) path integration distance, and (c) perspective taking error. Boxplots display group medians and interquartile ranges, with individual data points overlaid. Asterisks indicate levels of statistical significance (* $p < 0.05$; ** $p < 0.01$; *** $p < 0.001$).

c) Which method was used to conduct the post hoc tests? This relevant information is not reported.

The post hoc comparisons were conducted using the bootstrap-based robust method implemented in the Jamovi Robust ANOVA module (WRS2 R package). Specifically, post hoc tests were computed using the modified one-step M-estimator with 5,000 bootstrap resamples. This approach estimates pairwise group differences while maintaining robustness to heteroscedasticity and non-normality. The reported p-values are

derived from the empirical bootstrap distribution of the test statistic, and the accompanying 95% confidence intervals represent bias-corrected bootstrap intervals.

- d) **The post hoc comparisons are reported using p-values as if they indicated effect size. The authors should report effect sizes for each pairwise comparison and restructure this results section by including a qualitative interpretation of the effects' magnitude (e.g., small, moderate, large).**

We appreciate the reviewer's valuable comment. In response, we have conducted independent pairwise comparisons between CDR groups and computed robust effect sizes (ξ) for each contrast. These values are now reported in Supplementary Table 2, alongside the bootstrap confidence intervals. To provide a clearer quantitative interpretation, we have revised the Results section to include a summary of the magnitude of these effects across key contrasts. In addition, we have included a qualitative interpretation of the magnitude of the effects in the caption of Supplementary Table 2.

The updated paragraph in the *Results* section now reads as follows:

Paragraph added in the Results section: *Post-hoc comparisons revealed that participants with CDR 0 completed the training phase significantly faster than those with CDR 0.5 ($p = 0.022$, $\xi = 0.229$), CDR 1 ($p < 0.001$, $\xi = 0.717$), and CDR 2+ ($p < 0.001$, $\xi = 0.886$). The CDR 0.5 group was also significantly faster than both the CDR 1 ($p = 0.001$, $\xi = 0.419$) and CDR 2+ ($p < 0.001$, $\xi = 0.761$) groups. Finally, participants with CDR 1 were faster than those with CDR 2+ ($p = 0.022$, $\xi = 0.483$). For path integration distance error, participants with CDR 0 performed significantly better than those with CDR 1 ($p < 0.001$, $\xi = 0.666$), but not better than those with CDR 0.5 ($p = 0.440$). Participants with CDR 0.5 also outperformed those with CDR 1 ($p < 0.001$, $\xi = 0.714$). For perspective taking error, participants with CDR 0 performed significantly better than those with CDR 1 ($p < 0.001$, $\xi = 0.496$) and CDR 2+ ($p < 0.001$, $\xi = 0.661$) but did not differ from those with CDR 0.5 ($p = 0.572$). Participants with CDR 0.5 also outperformed those with CDR 1 ($p < 0.001$, $\xi = 0.518$) and CDR 2+ ($p < 0.001$, $\xi = 0.613$). No differences emerged between participants with CDR 1 and CDR 2+ ($p = 0.476$).*

- e) **Line 93-103: Redundant. The same information is already provided in the Statistical Analyses section.**

We thank the reviewer for noting this potential redundancy. However, the journal's format places the Methods section at the end of the manuscript. As such, we intentionally include slight repetition in the Results section to ensure that readers can readily understand the models being described without needing to refer back to the end of the paper.

4. **Discussion:** *As compared to the well-written introduction, the discussion is verbose and lacks breadth. In my opinion, the authors should emphasize the potential for such a serious game to increase patients' adherence to the assessment setting. Additionally, both the introduction and discussion lack theoretical references to egocentric and allocentric reference frames, which are specifically probed by the pointing task and the perspective-taking task, respectively. For reference purposes, please consult:*

Cavanna & Trimble. Brain. 2006. Doi:10.1093/brain/aw1004
Iardi et al. Aging Clin Exp Res. 2022. Doi:10.1007/s40520-021-01930-y
Ruggiero et al. Behav Brain Res. 2020. Doi:10.1016/j.bbr.2020.112793
Wood et al. Front Neurol. 2016. Doi:10.3389/fneur.2016.00215

We agree with the reviewer that the discussion would benefit from greater depth regarding the potential of serious games to enhance patient adherence to assessment and the theoretical foundations related to the reference frames

targeted by the tasks in SPACE. Concerning the latter, we have already addressed a similar point in our response to Reviewer 1 (Comment 6) and have revised the discussion accordingly.

References

- Arnold, A. E., Bures, F., Bray, S., Levy, R. M., & Iaria, G. (2014). Differential neural network configuration during human path integration. *Front Hum Neurosci*, 8, 263.
- Bai, W., Chen, P., Cai, H., Zhang, Q., Su, Z., Cheung, T., Jackson, T., Sha, S., & Xiang, Y.-T. (2022). Worldwide prevalence of mild cognitive impairment among community dwellers aged 50 years and older: a meta-analysis and systematic review of epidemiology studies. *Age and Ageing*, 51(8). <https://doi.org/10.1093/ageing/afac173>
- Berron, D., Glanz, W., Clark, L., Basche, K., Grande, X., Güsten, J., Billette, O. V., Hempen, I., Naveed, M. H., Diersch, N., Butryn, M., Spottke, A., Buerger, K., Perneczky, R., Schneider, A., Teipel, S., Wiltfang, J., Johnson, S., Wagner, M., ... Düzel, E. (2024). A remote digital memory composite to detect cognitive impairment in memory clinic samples in unsupervised settings using mobile devices. *NPJ Digital Medicine*, 7(1), 79.
- Bierbrauer, A., Kunz, L., Gomes, C. A., Luhmann, M., Deuker, L., Getzmann, S., Wascher, E., Gajewski, P. D., Hengstler, J. G., Fernandez-Alvarez, M., Atienza, M., Cammisuli, D. M., Bonatti, F., Pruneti, C., Percesepe, A., Bellaali, Y., Hanseeuw, B., Strange, B. A., Cantero, J. L., & Axmacher, N. (2020). Unmasking selective path integration deficits in Alzheimer's disease risk carriers. *Science Advances*, 6(35), eaba1394.
- Castegnaro, A., Ji, Z., Rudzka, K., Chan, D., & Burgess, N. (2023). Overestimation in angular path integration precedes Alzheimer's dementia. *Current Biology: CB*. <https://doi.org/10.1016/j.cub.2023.09.047>
- Cavanna, A. E., & Trimble, M. R. (2006). The precuneus: a review of its functional anatomy and behavioural correlates. *Brain: A Journal of Neurology*, 129(Pt 3), 564–583.
- Chan, D., Gallaher, L. M., Moodley, K., Minati, L., Burgess, N., & Hartley, T. (2016). The 4 Mountains Test: A Short Test of Spatial Memory with High Sensitivity for the Diagnosis of Pre-dementia Alzheimer's Disease. *Journal of Visualized Experiments: JoVE*, 116. <https://doi.org/10.3791/54454>
- Chong, J. R., Ashton, N. J., Karikari, T. K., Tanaka, T., Schöll, M., Zetterberg, H., Blennow, K., Chen, C. P., & Lai, M. K. P. (2021). Blood-based high sensitivity measurements of beta-amyloid and phosphorylated tau as biomarkers of Alzheimer's disease: a focused review on recent advances. *Journal of Neurology, Neurosurgery, and Psychiatry*, 92(11), 1231–1241.
- Chong, J. R., Hilal, S., Tan, B. Y., Venketasubramanian, N., Schöll, M., Zetterberg, H., Blennow, K., Ashton, N. J., Chen, C. P., & Lai, M. K. P. (2025). Clinical utility of plasma p-tau217 in identifying abnormal brain amyloid burden in an Asian cohort with high prevalence of concomitant cerebrovascular disease. *Alzheimer's & Dementia: The Journal of the Alzheimer's Association*, 21(2), e14502.
- Chrastil, E. R., Sherrill, K. R., Aselcioglu, I., Hasselmo, M. E., & Stern, C. E. (2017). Individual Differences in Human Path Integration Abilities Correlate with Gray Matter Volume in Retrosplenial Cortex, Hippocampus, and Medial Prefrontal Cortex. *ENeuro*, 4(2).
- Coughlan, G., Coutrot, A., Khondoker, M., Minihane, A.-M., Spiers, H., & Hornberger, M. (2019). Toward personalized cognitive diagnostics of at-genetic-risk Alzheimer's disease. *Proceedings of the National Academy of Sciences of the United States of America*, 116(19), 9285–9292.
- Coughlan, G., Laczó, J., Hort, J., Minihane, A.-M., & Hornberger, M. (2018). Spatial navigation deficits - overlooked cognitive marker for preclinical Alzheimer disease? *Nature Reviews. Neurology*, 14(8), 496–506.
- Coutrot, A., Silva, R., Manley, E., de Cothi, W., Sami, S., Bohbot, V. D., Wiener, J. M., Hölscher, C., Dalton, R. C., Hornberger, M., & Spiers, H. J. (2018). Global determinants of navigation ability. *Current Biology: CB*, 28(17), 2861-2866.e4.
- Dauphinot, V., Calvi, S., Moutet, C., Xie, J., Dautricourt, S., Batsavanis, A., Krolak-Salmon, P., & Garnier-Crussard, A. (2024). Reliability of the assessment of the clinical dementia rating scale from the analysis of medical records in comparison with the reference method. *Alzheimer's Research & Therapy*, 16(1), 198.
- Ekstrom, A. D., Arnold, A. E. G. F., & Iaria, G. (2014). A critical review of the allocentric spatial representation and its neural underpinnings: toward a network-based perspective. *Frontiers in Human Neuroscience*, 8, 803.

- Epstein, R. A., Patai, E. Z., Julian, J. B., & Spiers, H. J. (2017). The cognitive map in humans: spatial navigation and beyond. *Nature Neuroscience*, *20*(11), 1504–1513.
- Field, A. P., & Wilcox, R. R. (2017). Robust statistical methods: A primer for clinical psychology and experimental psychopathology researchers. *Behaviour Research and Therapy*, *98*, 19–38.
- Hamari, J., Koivisto, J., & Sarsa, H. (2014). Does gamification work?—a literature review of empirical studies on gamification. In *47th Hawaii international conference on system sciences* (pp. 3025–3034). Ieee.
- Hampel, H., Au, R., Matke, S., van der Flier, W. M., Aisen, P., Apostolova, L., Chen, C., Cho, M., De Santi, S., Gao, P., Iwata, A., Kurzman, R., Saykin, A. J., Teipel, S., Vellas, B., Vergallo, A., Wang, H., & Cummings, J. (2022). Designing the next-generation clinical care pathway for Alzheimer’s disease. *Nature Aging*, *2*(8), 692–703.
- Hilal, S., Chai, Y. L., Ikram, M. K., Elangovan, S., Yeow, T. B., Xin, X., Chong, J. Y., Venketasubramanian, N., Richards, A. M., Chong, J. P. C., Lai, M. K. P., & Chen, C. (2015). Markers of cardiac dysfunction in cognitive impairment and dementia. *Medicine*, *94*(1), e297.
- Howett, D., Castegnaro, A., Krzywicka, K., Hagman, J., Marchment, D., Henson, R., Rio, M., King, J. A., Burgess, N., & Chan, D. (2019). Differentiation of mild cognitive impairment using an entorhinal cortex-based test of virtual reality navigation. *Brain: A Journal of Neurology*, *142*(6), 1751–1766.
- Iaria, G., Petrides, M., Dagher, A., Pike, B., & Bohbot, V. D. (2003). Cognitive strategies dependent on the hippocampus and caudate nucleus in human navigation: variability and change with practice. *J Neurosci*, *23*(13), 5945–5952.
- Ilardi, C. R., Chieffi, S., Iachini, T., & Iavarone, A. (2022). Neuropsychology of posteromedial parietal cortex and conversion factors from Mild Cognitive Impairment to Alzheimer’s disease: systematic search and state-of-the-art review. *Aging Clinical and Experimental Research*, *34*(2), 289–307.
- Ilardi, C. R., Menichelli, A., Michelutti, M., Cattaruzza, T., Federico, G., Salvatore, M., Iavarone, A., & Manganotti, P. (2024). On the clinimetrics of the Montreal Cognitive Assessment: Cutoff analysis in patients with mild cognitive impairment due to Alzheimer’s disease. *Journal of Alzheimer’s Disease: JAD*, *101*(1), 293–308.
- Jack, C. R., Jr, Andrews, J. S., Beach, T. G., Buracchio, T., Dunn, B., Graf, A., Hansson, O., Ho, C., Jagust, W., McDade, E., Molinuevo, J. L., Okonkwo, O. C., Pani, L., Rafii, M. S., Scheltens, P., Siemers, E., Snyder, H. M., Sperling, R., Teunissen, C. E., & Carrillo, M. C. (2024). Revised criteria for diagnosis and staging of Alzheimer’s disease: Alzheimer’s Association Workgroup. *Alzheimer’s & Dementia: The Journal of the Alzheimer’s Association*, *20*(8), 5143–5169.
- Kozhevnikov, M., & Hegarty, M. (2001). A dissociation between object manipulation spatial ability and spatial orientation ability. *Memory & Cognition*, *29*(5), 745–756.
- Lambrey, S., Doeller, C., Berthoz, A., & Burgess, N. (2012). Imagining being somewhere else: neural basis of changing perspective in space. *Cerebral Cortex (New York, N.Y.: 1991)*, *22*(1), 166–174.
- Liew, T. M., Foo, J. Y. H., Yang, H., Tay, S. Y., Koay, W. I., Yip, K. F., Ting, S. K. S., Narasimhalu, K., Li, W., Tan, C., Luo, D., Chong, R., Shong, R., Sia, C., Koh, G. C.-H., & Thumboo, J. (2025). PENSIEVE-AI a brief cognitive test to detect cognitive impairment across diverse literacy. *Nature Communications*, *16*(1), 2847.
- Lim, M. J. H., Cheung, C. Y., Chong, J. R., Chua, J., Hilal, S., Lai, M. K. P., Maier, A. B., Schmetterer, L., Tan, B. Y., Venketasubramanian, N., Wong, T. Y., Xu, X., Yeo, B. T. T., Zhou, J. H., & Chen, C. L. H. (2025). HARMONISATION – A multimodal prospective study of vascular cognitive impairment in multi-ethnic Asians: Cohort profile, progress, current contributions, and future impact. *Journal of Alzheimer’s Disease*, *0*(0). <https://doi.org/10.1177/13872877251389006>
- Mair, P., & Wilcox, R. (2020). Robust statistical methods in R using the WRS2 package. *Behavior Research Methods*, *52*(2), 464–488.
- Mcalearney, A. S. (2016). High Touch and High Tech (HT2) proposal: transforming patient engagement throughout the continuum of care by engaging patients with portal technology at the bedside. *JMIR Res. Protoc*, *5*.
- Meier, I. B., Buegler, M., Harms, R., Seixas, A., Çöltekin, A., & Tarnanas, I. (2021). Using a Digital Neuro Signature to measure longitudinal individual-level change in Alzheimer’s disease: the Altoida large cohort study. *Npj Digital Medicine*, *4*(1).

- Mokrisova, I., Laczó, J., Andel, R., Gazova, I., Vyhnaček, M., Nedelska, Z., Levcik, D., Cerman, J., Vlček, K., & Hort, J. (2016). Real-space path integration is impaired in Alzheimer's disease and mild cognitive impairment. *Behavioural Brain Research*, 307, 150–158.
- Morris, J. C. (1993). The Clinical Dementia Rating (CDR): current version and scoring rules: Current version and scoring rules. *Neurology*, 43(11), 2412–2414.
- Newton, C., Pope, M., Rua, C., Henson, R., Ji, Z., Burgess, N., & Others. (2024). Entorhinal-based path integration selectively predicts midlife risk of Alzheimer's disease. *Alzheimers Dement*, 20(4), 2779–2793.
- Polk, S. E., Öhman, F., Hassenstab, J., König, A., Papp, K. V., Schöll, M., & Berron, D. (2025). A scoping review of remote and unsupervised digital cognitive assessments in preclinical Alzheimer's disease. *Npj Digital Medicine*, 8(1), 266.
- Rekers, S., & Finke, C. (2024). Translating spatial navigation evaluation from experimental to clinical settings: The virtual environments navigation assessment (VIENNA). *Behavior Research Methods*, 56(3), 2033–2048.
- Rentz, D. M., Parra Rodriguez, M. A., Amariglio, R., Stern, Y., Sperling, R., & Ferris, S. (2013). Promising developments in neuropsychological approaches for the detection of preclinical Alzheimer's disease: a selective review. *Alzheimer's Research & Therapy*, 5(6), 58.
- Ruggiero, G., Ruotolo, F., Iavarone, A., & Iachini, T. (2020). Allocentric coordinate spatial representations are impaired in aMCI and Alzheimer's disease patients. *Behavioural Brain Research*, 393(112793), 112793.
- Salari, N., Lotfi, F., Abdolmaleki, A., Heidarian, P., Rasoulpoor, S., Fazeli, J., Najafi, H., & Mohammadi, M. (2025). The global prevalence of mild cognitive impairment in geriatric population with emphasis on influential factors: a systematic review and meta-analysis. *BMC Geriatrics*, 25(1), 313.
- Schinazi, V. R., Nardi, D., Newcombe, N. S., Shipley, T. F., & Epstein, R. A. (2013). Hippocampal size predicts rapid learning of a cognitive map in humans. *Hippocampus*, 23(6), 515–528.
- Schinazi, V. R., & Thrash, T. (2018). Cognitive neuroscience of spatial and geographic thinking. In *Handbook of Behavioral and Cognitive Geography*. Edward Elgar Publishing.
- Segen, V., Ying, J., Morgan, E., Brandon, M., & Wolbers, T. (2022). Path integration in normal aging and Alzheimer's disease. *Trends in Cognitive Sciences*, 26(2), 142–158.
- Tham, W., Auchus, A. P., Thong, M., Goh, M.-L., Chang, H.-M., Wong, M.-C., & Chen, C. P. L.-H. (2002). Progression of cognitive impairment after stroke: one year results from a longitudinal study of Singaporean stroke patients. *Journal of the Neurological Sciences*, 203–204, 49–52.
- Tu, S., Wong, S., Hodges, J. R., Irish, M., Pigué, O., & Hornberger, M. (2015). Lost in spatial translation - A novel tool to objectively assess spatial disorientation in Alzheimer's disease and frontotemporal dementia. *Cortex; a Journal Devoted to the Study of the Nervous System and Behavior*, 67, 83–94.
- van der Ham, I. J. M., Claessen, M. H. G., Evers, A. W. M., & van der Kuil, M. N. A. (2020). Large-scale assessment of human navigation ability across the lifespan. *Scientific Reports*, 10(1), 3299.
- Watermeyer, T., & Calia, C. (2019). Neuropsychological assessment in preclinical and prodromal Alzheimer disease: a global perspective. *Journal of Global Health*, 9(1), 010317.
- Weisberg, S. M., Schinazi, V. R., Newcombe, N. S., Shipley, T. F., & Epstein, R. A. (2014). Variations in cognitive maps: Understanding individual differences in navigation. *Journal of Experimental Psychology. Learning, Memory, and Cognition*, 40(3), 669–682.
- Wolbers, T., & Buchel, C. (2005). Dissociable retrosplenial and hippocampal contributions to successful formation of survey representations. *J Neurosci*, 25(13), 3333–3340.
- Wood, R. A., Moodley, K. K., Lever, C., Minati, L., & Chan, D. (2016). Allocentric spatial memory testing predicts conversion from mild cognitive impairment to dementia: An initial proof-of-concept study. *Frontiers in Neurology*, 7, 215.
- Xu, X., Chew, K. A., Wong, Z. X., Phua, A. K. S., Chong, E. J. Y., Teo, C. K. L., Sathe, N., Chooi, Y. C., Chia, W. P. F., Henry, C. J., Chew, E., Wang, M., Maier, A. B., Kandiah, N., & Chen, C. L.-H. (2022). The SINGapore GERiatric intervention

study to reduce cognitive decline and physical frailty (SINGER): Study design and protocol. *The Journal of Prevention of Alzheimer's Disease*, 9(1), 40–48.